# Modular Deep Learning

**Jonas Pfeiffer***  *jonaspfeiffer@google.com*
*Google DeepMind*

**Sebastian Ruder***  *ruder@google.com*
*Google DeepMind*

**Ivan Vulić**  *iv250@cam.ac.uk*
*University of Cambridge*

**Edoardo M. Ponti***  *eponti@ed.ac.uk*
*University of Edinburgh*
*University of Cambridge*

**Reviewed on OpenReview:** `https://openreview.net/forum?id=z9EkXfvxta`

## Abstract

Transfer learning has recently become the dominant paradigm of machine learning. Pre-trained models fine-tuned for downstream tasks achieve better performance with fewer labelled examples. Nonetheless, it remains unclear how to develop models that specialise towards multiple tasks without incurring negative interference and that generalise systematically to non-identically distributed tasks. Modular deep learning has emerged as a promising solution to these challenges. In this framework, units of computation are often implemented as autonomous parameter-efficient modules. Information is conditionally routed to a subset of modules and subsequently aggregated. These properties enable positive transfer and systematic generalisation by separating computation from routing and updating modules locally. We offer a survey of modular architectures, providing a unified view over several threads of research that evolved independently in the scientific literature. Moreover, we explore various additional purposes of modularity, including scaling language models, causal inference and discovery, programme simulation, and hierarchical reinforcement learning. Finally, we report various concrete applications where modularity has been successfully deployed such as cross-lingual and cross-modal knowledge transfer.

## Table of Contents

---

*Authors contributed equally.

# 1 Introduction and Motivation

Transfer learning has recently become pervasive in machine learning technology, such as in natural language processing (Ruder et al., 2019b; Brown et al., 2020), computer vision (Dosovitskiy et al., 2021), and reinforcement learning (Reed et al., 2022), among other areas. In its most successful incarnation, transfer learning consists of pre-training a model on vast amounts of raw data in a self-supervised fashion and subsequently fine-tuning it for new tasks based on a small number of labelled examples. Despite its success, this paradigm for transfer learning suffers from a series of limitations in various settings. Firstly, in multi-task fine-tuning, the learning signals from different tasks may *negatively interfere* with each other (McCloskey & Cohen, 1989). Similarly, in continuous learning, adapting to new examples can result in *catastrophic forgetting* of knowledge acquired from previous examples (Sutton, 1986; French, 1999).[1] Secondly, in settings where the training and evaluation distributions are not identical, these models fail in *generalising systematically* (Lake & Baroni, 2018; Hupkes et al., 2020). This makes models brittle and inaccurate and hampers their deployment in real-world applications, where distribution shifts are common.

In contrast, many biological and artificial systems do not suffer from these weaknesses by virtue of their *modularity* (Fodor, 1983; Ballard, 1986), defined as the correspondence between strongly interconnected components of a system (i.e., modules) and the functions they perform (Baldwin & Clark, 2000; Ulrich, 1995). In other words, each module is *specialised* for a unique purpose, for which it is reused consistently. In animal brains, this favours *evolvability*, the ability to adapt quickly to new environments, and *resilience* to environment perturbations (Wagner et al., 2005) because it makes rewiring connections easier than in monolithic, entangled networks (Kashtan & Alon, 2005). Artificial systems, such as programming languages and computer hardware, are similarly designed in a modular fashion (Booch et al., 2008; Baldwin & Clark, 2000) because this modular design favours consistency, ease of adaptation, and interpretability.

To what extent, then, do 'vanilla' neural networks display the desirable property of being modular? In principle, given their fully connected nature, they could develop such a structure as a by-product of optimising a loss for a downstream task. Recent structural analyses based on hierarchical clustering of neurons revealed that vanilla neural networks can indeed learn such a modular pattern (Watanabe, 2019; Casper et al., 2022; Foroutan et al., 2022). Favourable conditions for the emergence of modularity include multi-task learning (Dobs et al., 2022) and regularisation through dropout (Lange et al., 2022). In particular, from a structural perspective, populations of neurons may activate jointly in response to specific features of the input or the output classes[2], resulting in similar changes in model performance when ablated (Meyes et al., 2020). From a functional perspective, multi-task learning may lead to segregated, specialised sub-networks (Yang et al., 2019; Dobs et al., 2022). On the other hand, Csordás et al. (2021) revealed that a given sub-network does not tend to be re-used for similar sub-tasks nor to be combined with others to express more complex functions. In fact, in many cases, the performance of a model on simple tasks requiring a certain skill and composite tasks requiring a combination thereof is entirely uncorrelated (Li et al., 2022a).

For this reason, previous work explored the idea of designing neural networks that are *explicitly* modular (Jacobs et al., 1991a; Rosenbaum et al., 2018; Ponti, 2021; Mittal et al., 2022). This has the goal of achieving not only *functional specialisation* (Zhang et al., 2022b), but also *re-usability* and *composability*. In particular, these methods involve identifying 1) *modules* in a neural network that can be updated locally and asynchronously, without affecting the rest of the parameters; 2) a *routing function* that chooses a subset of modules for each example or task; and 3) an *aggregation function* that aggregates the outputs of the active modules. Each of these three ingredients can be manually specified or learned. We provide several case studies of different configurations of these components in Figure 1.

The main advantages of modular neural architectures are *positive transfer*, *compositionality*, and *parameter efficiency*. Firstly, modularity encourages positive transfer by encoding similar functions with the same module. At the same time, it prevents interference and forgetting by allocating distinct functions to different dedicated modules (Jacobs et al., 1991b). For instance, massively multilingual Transformer-based models in NLP are

---

[1]These phenomena have also been referred to as spatial and temporal 'crosstalk' (Jacobs et al., 1991b).

[2]Lange et al. (2022) found that clusters identified through downstream (output) information do not match with the clusters identified through upstream (input) information. They attribute this phenomenon to their different roles, namely disentanglement of the input structure and composition of the output structure, respectively.

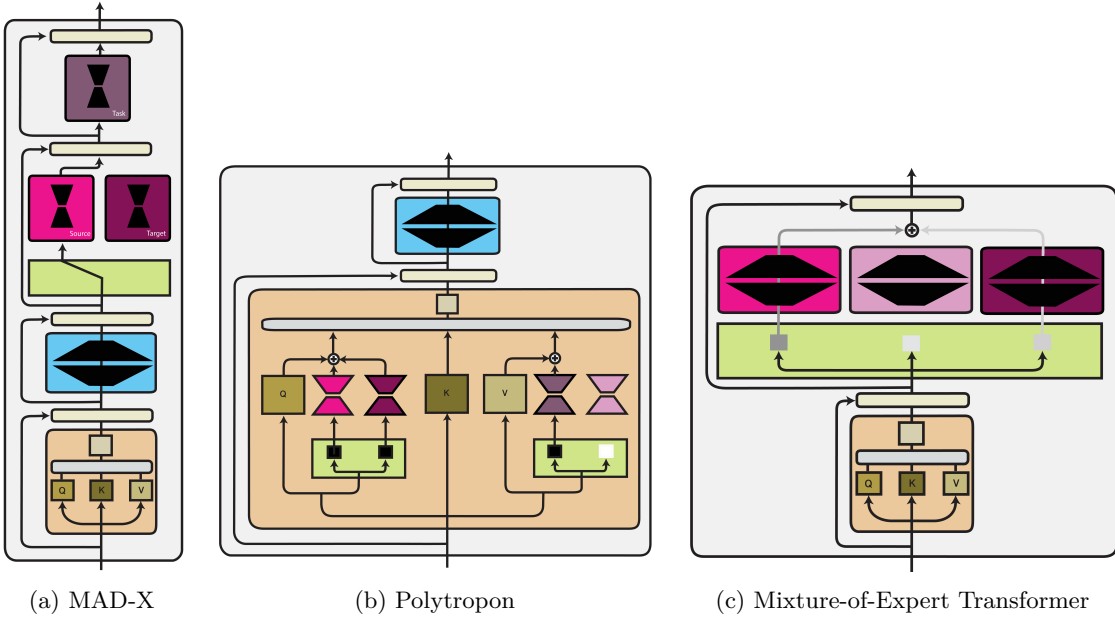

(a) MAD-X       (b) Polytropon       (c) Mixture-of-Expert Transformer

Figure 1: Case studies of modular deep learning; best viewed in colour. Green components illustrate different routing functions (see § 4); shade-of-purple components illustrate modular computation functions (see §3). 1a) MAD-X (Pfeiffer et al., 2020b) uses Adapter layers with fixed routing for zero-shot cross-lingual transfer. 1b) Polytropon (Ponti et al., 2022) uses low-rank adapters (LoRA; Hu et al., 2022) with hard learned routing for few-shot task adaptation. 1c) MoE Transformers (Fedus et al., 2021; Clark et al., 2022, *inter alia*) use Multi-Layer Perceptrons with top-$k$ soft routing, in order to scale to larger model sizes. The three representative models illustrated here are only a fraction of possible configurations from the 'configuration manifold' that can be created by varying the components surveyed in §3-§6.

known to suffer from a 'curse of multilinguality' (Conneau et al., 2020) due to the conflicting information that the gradient from each language-specific loss carries (Wang et al., 2021b). A possible solution is augmenting these entangled, fully shared models with specialised modules responsible for individual languages (Pfeiffer et al., 2020b; 2022b). More generally, as the range of tasks modelled jointly by a single model becomes increasingly diverse, modularity may be instrumental in the advent of general-purpose, multi-modal agents that encompass vision, language, and action (Reed et al., 2022).

Secondly, modules representing different skills (at the task level) or features (at the example level) can be composed together and updated locally, without affecting the rest of the network. These two properties are crucial in two main settings, which correspond to different aspects of *systematic generalisation*: one is the ability to *re-compose*, i.e. zero-shot transfer to tasks consisting of new combinations of learned skills, or examples consisting of new combinations of observed features (Hupkes et al., 2020). For instance, while modules for the Guaraní language and for dependency parsing can only be trained separately due to the lack of annotated data for dependency parsing in Guaraní, they can be composed to perform inference on this unobserved task–language combination (Pfeiffer et al., 2020b). Similarly, in hierarchical reinforcement learning, an agent can follow different sequences of modular policies known as options in tasks requiring the completion of similar sub-goals in different orders (Sutton et al., 1999; Precup, 2000). The other aspect of systematic generalisation is *robustness*. In fact, if modules are taken to correspond to independent and reusable physical mechanisms (Schölkopf et al., 2012), *local* shifts in their distributions require updating only the parameters accounting for the affected skills or features (Goyal et al., 2021; Schölkopf et al., 2021), while the rest of the model remains invariant to the change. In practice, the ability to perform local updates facilitates sample efficiency, as fewer examples are necessary to adapt models to new tasks (Bengio et al., 2020; Ponti et al., 2022).

Thirdly, an additional advantage of modular neural architectures is parameter and time *efficiency*. In this framework, fine-tuning a model on a specific task only requires storing a modular adapter rather than a separate copy of the entire (typically large) model. What is more, modules can be added or removed on-the-fly in an incremental manner, adjusting the model capacity according to the task complexity. This ability is known as *conditional computation* (Bengio et al., 2015). Finally, modularity enables language models to scale to larger numbers of parameters while retaining the same time complexity, by selecting only a small set of experts per example (Shazeer et al., 2017; Fedus et al., 2021).

As the main contribution of this survey, we offer a unified view of modular deep learning, illustrating how many families of methods can be defined along four key dimensions: **1)** how they implement modules, which constitute the minimum unit of computation; **2)** how they select active modules through a routing function; **3)** how module outputs are aggregated; and **4)** how the modules are trained with the rest of the model.

For module implementation, we discuss sparse subnetworks (Hu et al., 2022; Ansell et al., 2022), adapter layers (Rebuffi et al., 2018; Pfeiffer et al., 2020b), and prefix tuning (Li & Liang, 2021), among others. These methods have been proven as an effective way to adapt large pre-trained models, achieving better performance and sample efficiency than alternative strategies such as in-context learning (Liu et al., 2022b), which may be brittle (Lu et al., 2022). In fact, modules can also take the form of human-engineered prompts, where the model is provided with input–output examples (Brown et al., 2020) or task instructions (Wei et al., 2022a). While many module implementations share the same underlying functional form (He et al., 2021), they offer different trade-offs between efficiency and performance.

We then discuss how routing functions control the flow of information to the modules: in fixed routing, module allocation is manually defined when expert knowledge is available(Hampshire & Waibel, 1992; Rajendran et al., 2017, *inter alia*). In learned routing, a parameterised routing function is inferred during training. This, however, poses a series of challenges, such as training instability, module collapse, and overfitting (Rosenbaum et al., 2019). Orthogonally, we also distinguish between hard and soft routing. In hard routing, only a subset of modules is activated (Rosenbaum et al., 2018; Ponti et al., 2022; Fernando et al., 2017, *inter alia*). In soft routing, all modules are aggregated according to continuous scores (Jacobs et al., 1991b; Jordan & Jacobs, 1994). While soft routing is amenable to vanilla gradient descent, it is highly inefficient. On the other hand, hard routing requires approximate inference but facilitates conditional computation and module specialisation. When multiple modules are selected, several *aggregation* strategies are possible. For instance, these can be based on interpolating the parameters of active modules (Ansell et al., 2022) or an attention mechanism over the module outputs (Pfeiffer et al., 2021a). Alternative methods include input prompt concatenation (Vu et al., 2022b) and function composition (Andreas et al., 2016b).

Finally, modules can be trained jointly with the rest of the base model in multi-task learning (Caruana, 1997; Ruder, 2017), added sequentially in classic continual learning (Rusu et al., 2016), or integrated post-hoc into an already pre-trained and frozen model (Rebuffi et al., 2017; Houlsby et al., 2019). The last scenario is most common with current state-of-the-art models, which are trained as dense, fully shared models and may be 'modularised' after pre-training.

Crucially, this taxonomy reveals unexpected connections between several independent threads of research, including aggregation functions and mode connectivity (Frankle et al., 2020), routing and hypernetworks (Ha et al., 2017), among others. We further illustrate a series of applications of modular networks in *transfer learning* across different areas such as natural language processing, computer vision, and speech processing. In addition, we show how modularity plays an important role in causal inference and discovery, programme simulation, and hierarchical reinforcement learning. We hope that our overview will spark future research on modular deep learning in areas that may benefit from it such as community-driven efforts to develop and maintain machine learning technology.

## 2 Modular Deep Learning

This survey focuses on modular deep learning: namely, on models composed of modules. These are autonomous computation functions that, depending on their architecture and purpose, are variously referred to as adapters (Rebuffi et al., 2017; Pfeiffer et al., 2020a), options (Sutton et al., 1999; Precup, 2000), or experts (Jacobs

| Notation | Definition |
|---|---|
| $\boldsymbol{x} \in \mathcal{X}$ | Input data |
| $\boldsymbol{y} \in \mathcal{Y}$ | Output data |
| $\boldsymbol{h} \in \mathcal{H}$ | Hidden representation |
| $t \in \mathcal{T}$ | Task index |
| $f : \mathcal{X} \cup \mathcal{H} \to \mathcal{Y} \cup \mathcal{H}$ | A computation function |
| $\boldsymbol{\theta}$ | Shared parameters |
| $M = \{\boldsymbol{\phi}_1, \ldots, \boldsymbol{\phi}_{|M|}\}$ | Set of module parameters |
| $\boldsymbol{\alpha} \in \mathcal{A}$ | Vector of routing scores |
| $r : \mathcal{X} \cup \mathcal{H} \cup \mathcal{T} \to \mathcal{A}$ | Routing function |
| $\boldsymbol{\rho}$ | Routing parameters |
| $g$ | Aggregation function |
| $\boldsymbol{\gamma}$ | Aggregation parameters |

Table 1: Notation and definition of important variables, functions, and operators.

et al., 1991a; Jordan & Jacobs, 1994). Crucially, these modules are distinguished from a routing function, which controls the information flow to the modules. Finally, an aggregation function aggregates their outputs. Modules can be optionally combined with fully shared (thus, non-modular) parameters as part of the same neural architecture. In order to provide a unified view of the landscape of modular deep learning, we create a taxonomy of four dimensions of variation: computation, routing, aggregation, and training. These dimensions are mutually independent; hence, many methods can be interpreted as different combinations of these dimensions, listed in § 2.1. Concurrently, we provide a unified, consistent notation in § 2.2, which helps illuminate the relationship among such methods.

### 2.1 Taxonomy

**1) Computation function**: *How is each module implemented?* (§ 3) A module may consist of any component of a neural architecture, such as multiple copies of a model (Jacobs et al., 1991a) or one of its layers (Fedus et al., 2021). Alternatively, as it is common in transfer learning, modules can be combined with a function parameterised by fully shared pre-trained weights. In this case, we distinguish between modification of parameters (parameter composition), concatenation with input features (input composition), and function composition by stacking neural modules.

**2) Routing function:** *How are active modules selected?* (§ 4) Under *fixed routing*, we categorise approaches where the routing function is fixed. This assumes that the specialisation of each module, as well as the combination of modules required for each task, is known *a priori*. In *learned routing*, the parameters of the routing mechanism are learned during training. In this case, routing is soft if all modules are ranked through a continuous score, or hard if each module is given a binary score (active or inactive).

**3) Aggregation function:** *How are the outputs of the active modules aggregated?* (§ 5) We differentiate between methods that compose the outputs of the active modules deterministically (e.g., based on a weighted average) from those where the aggregation function is implemented as a learnable neural network that depends on the output of all modules.

**4) Training setting:** *How are the modules trained?* (§ 6) Some methods, such as MoEs, train the modules (and possibly the routing function) jointly with the shared weights of a randomly initialised model. As an alternative, transfer learning approaches introduce modules *post-hoc* after pre-training weights and adapt them during fine-tuning. In continuous learning settings, instead, new modules may be introduced iteratively for every new task in a sequence.

## 2.2 Notation

More formally, let a neural network $f_{\boldsymbol{\theta}} : \mathcal{X} \rightarrow \mathcal{Y}$ be decomposed into a graph of sub-functions. In the simplest case, this graph is a linear chain $f_{\boldsymbol{\theta}_1} \circ f_{\boldsymbol{\theta}_2} \circ \cdots \circ f_{\boldsymbol{\theta}_l}$, where $\circ$ stands for function composition. These sub-functions refer to the model's $l$ layers, each with unique indexed parameters $\boldsymbol{\theta}_i, i = 1, \ldots, l$.[3] In turn, these can be further decomposed recursively into a graph of their constituent sub-functions: for instance, a Transformer layer (Vaswani et al., 2017) includes linear mappings for the query, key, value, and output, as well as a non-linear feed-forward network, and residual connections. We further denote the values of the parameters at initialisation as $\boldsymbol{\theta}^0$, and the parameters after training are denoted as $\boldsymbol{\theta}^\star$.

Any $i$-th sub-function with input $\boldsymbol{x}$ can be modified by a module with parameters $\boldsymbol{\phi}$ from the inventory $M_i = f_{\boldsymbol{\phi}_1}, \ldots, f_{\boldsymbol{\phi}_{|M|}}$ in the following different ways:

1. *parameter composition*: $f'_i(\boldsymbol{x}) = f_{\boldsymbol{\theta}_i \oplus \boldsymbol{\phi}}(\boldsymbol{x})$, where $\oplus$ stands for an operation that composes the original parameters with the module parameters, such as element-wise addition. An example is low-rank (Hu et al., 2022) or sparse (Ansell et al., 2022) adapters.

2. *input composition*: $f'_i(\boldsymbol{x}) = f_{\boldsymbol{\theta}_i}([\boldsymbol{\phi}, \boldsymbol{x}])$, where $[\cdot, \cdot]$ stands for concatenation. An example is prefix tuning Li & Liang (2021).

3. *function composition*: $f'_i(\boldsymbol{x}) = f_{\boldsymbol{\phi}} \circ f_{\boldsymbol{\theta}_i}(\boldsymbol{x})$, where the outputs of the first function is fed into the second function. An example are adapter layers (Rebuffi et al., 2017).

For each $i$-th sub-function, multiple modules from the inventory $M_i$ can be selected through a routing function $r(\cdot)$, which returns a score $\alpha_j$ for each module $f_{\boldsymbol{\phi}_j}$ conditioned on the data itself, such as a language token or visual region $x$ or the full input $\boldsymbol{x}$, or metadata such as the task identity $t \in \mathcal{T}$. Note that $\boldsymbol{\alpha}$ can be fixed *a priori* through expert knowledge or learned through an appropriate parameterisation $r_{\boldsymbol{\rho}}(\cdot)$, where $\boldsymbol{\rho}$ refers to (learnable) parameters of the routing function. Often, the routing function takes special forms:

1. In *hard* routing, $\boldsymbol{\alpha} \in \{0, 1\}^{|M|}$ is a discrete binary vector. If these parameters are learned, inference usually relies on score function estimators, stochastic re-parameterisation, or evolutionary algorithms.

2. In *soft* routing, $\boldsymbol{\alpha} \in [0, 1]^{|M|}$ is a continuous probability distribution, such that $\sum_j \alpha_j = 1$.

3. Finally, $\boldsymbol{\alpha} \in \mathbb{R}^{|M|}$ can be an unnormalised score vector. This is the case in linear *hypernetworks* (Ha et al., 2017), where $\boldsymbol{\alpha}$ is usually interpreted as a task embedding and the row-wise stacked module parameters $\Phi = [\boldsymbol{\phi}_1, \ldots, \boldsymbol{\phi}_{|M|}]$ act as a parameter generator.

Finally, the output of each module is combined through an aggregation function $g(\cdot)$.[4] The aggregation function usually takes two possible forms. One consists of a deterministic operation based on the routing scores (e.g., weighted averaging of module parameters or outputs). The other consists of a learnable neural network, such as an attention mechanism between the modules' inputs and outputs (Pfeiffer et al., 2021a). When we put the computation function, routing function, and aggregation function together, we obtain the general recipe for a modular function, illustrated in Algorithm 1.

---

[3]We abuse notation by treating indexing over functions, $f_i$, as identical to indexing over the parameters of a function, $f_{\boldsymbol{\theta}_i}$. In this survey, both are used interchangeably.

[4]To avoid clutter in terminology, throughout this work we use the term *composition* to refer to the merger of computation functions (§ 3), and the term *aggregation* to refer to different approaches of combining the outputs of different modules (§ 5).

---
**Algorithm 1:** Forward pass of a modular function

---
**1** Inputs: example $\boldsymbol{x}$, task $t$
**2** $\boldsymbol{\alpha} \leftarrow r_{\boldsymbol{\rho}}(\boldsymbol{x}, t)$ // Routing
**3** $H \leftarrow \{\}$
**4** **for** $\boldsymbol{\phi}_j \in M_i$ **do**
**5**    $\boldsymbol{h}_j \leftarrow f(\boldsymbol{x}; \boldsymbol{\theta}_i, \boldsymbol{\phi}_j)$ // Computation
**6**    $H \leftarrow H \cup \boldsymbol{h}_j$
**7** $\boldsymbol{y} \leftarrow g_{\gamma}(\boldsymbol{\alpha}, H)$ // Aggregation

---

Given shared parameters $\boldsymbol{\theta}_i$ for the $i$-th sub-function and a corresponding inventory of modules $M_i$, we first sample a task $t$, and an input $\boldsymbol{x}$. The routing scores $\boldsymbol{\alpha}$ are obtained from the routing function $r(\cdot)$. We now compute the hidden representation $\boldsymbol{h}_j$ of each module $\boldsymbol{\phi}_j$ and aggregate them with the function $g(\cdot)$ into the output $\boldsymbol{y}$. We elaborate on the settings for training these different components in § 6. We provide an overview of representative computation, routing, and aggregation functions in Table 2.

## 3 Computation Function

The computation function determines the design of a module. Various module architectures have been proposed such as MLP layers (Rosenbaum et al., 2018; Kirsch et al., 2018; Chang et al., 2019), independent RNNs (Goyal et al., 2021), independent CNNs (Parascandolo et al., 2018), or special-purpose architectures (Andreas et al., 2016b). However, in transfer learning, modules are most often integrated into a base architecture whose parameters are fully shared. We identify three core methods to merge a *single* module with the corresponding sub-function: parameter composition, input composition, and function composition. While all three methods instantiate modules differently, we demonstrate how they can be seen in a unified view in § 3.5. We provide example illustrations of the three computation functions (in addition to a hypernetwork) as part of a Transformer architecture in Figure 2 and provide a high-level overview of their trade-offs in Table 3, which we further discuss in the respective sections.[5]

### 3.1 Parameter Composition

Parameter composition methods augment the function $f_W$ of a base model with weights $W \in \mathbb{R}^{o \times i}$ with module parameters $\Phi \in \mathbb{R}^{o \times i}$, where $i$ is the input dimensionality, and $o$ is the output dimensionality. In particular, the module inventory consists of a set of sparse or low-rank weights to ensure that the modules are parameter-efficient. Therefore, the resulting function is parameterised as $f_{\boldsymbol{\theta} \oplus \boldsymbol{\phi}_i}$, where $\oplus$ stands for element-wise addition.

**Sparse Subnetworks**    *Sparsity* is a common inductive bias based on the assumptions (i) that only a small number of parameters of an over-parameterised model are relevant for a particular task, and that (ii) similar tasks share similar sub-networks. This is the case, for instance, for language subnetworks in multilingual language models (Stanczak et al., 2022; Foroutan et al., 2022).

The most widespread method to induce sparsity is *pruning*. This can be interpreted as the application of a binary mask $\boldsymbol{b} \in \{0, 1\}^{|\boldsymbol{\theta}|}$ that selectively keeps or removes each connection in a model with trained parameters $\boldsymbol{\theta}^{\star}$: $f' = f_{\boldsymbol{\theta}^{\star} \odot \boldsymbol{b}}$ where $\odot$ is element-wise multiplication. The merger of $\boldsymbol{\theta}$ and $\boldsymbol{b}$ results in a sparse subnetwork, but the corresponding model parameters usually remain dense for hardware and software reasons.[6] After training, the trained weights are sorted based on a criterion and a fraction (bottom-$k$) of the weights are set to zero. Examples of criteria include magnitude after convergence (Han et al., 2017) and change of magnitude between initialisation and convergence (Frankle & Carbin, 2019).

---

[5]The comparison is mainly meant as a high-level guideline. Individual methods may have different trade-offs and mitigate certain weaknesses indicated in the table.

[6]In fact, sparse linear algebra operations on graphic processing units remain highly inefficient, if available at all. Examples include the sparse tensor classes in Pytorch: https://pytorch.org/docs/stable/sparse.html

| | Method | Reference | Function |
|---|---|---|---|
| **Computation function** | Sparse subnetwork | Frankle & Carbin (2019) | $f' = f_{\boldsymbol{\theta}^\star \odot \boldsymbol{b}}$ |
| | Supermasks | Wortsman et al. (2020) | $f' = f_{\boldsymbol{\theta}_0 \odot \boldsymbol{b}}$ |
| | Sparse fine-tuning | Ansell et al. (2022) | $f' = f_{\boldsymbol{\theta} + \boldsymbol{b} \odot \boldsymbol{\phi}}$ |
| | Intrinsic dimension | Li et al. (2018) | $f' = f_{\boldsymbol{\theta} + \boldsymbol{\phi}\mathbf{M}}$ |
| | Low-rank adaptation | Hu et al. (2022) | $f_i' = f_{\boldsymbol{\theta}_i + \mathrm{vec}(\boldsymbol{B}_i \boldsymbol{A}_i)}$ |
| | Prompting | Brown et al. (2020) | $f_1' = f_{\boldsymbol{\theta}_1}([\boldsymbol{\phi}, \boldsymbol{x}])$ where $\boldsymbol{\phi} = \mathrm{Emb}(\boldsymbol{p})$ |
| | Retrieval augmentation | Guu et al. (2020) | $f_1' = f_{\boldsymbol{\theta}_1}([\boldsymbol{\phi}, \boldsymbol{x}])$ where $\boldsymbol{\phi} = \mathrm{Emb}([\boldsymbol{p}, \boldsymbol{c}])$ |
| | Prompt tuning | Lester et al. (2021) | $f_1' = f_{\boldsymbol{\theta}_1}([\boldsymbol{\phi}, \boldsymbol{x}])$ |
| | Multi-layer prompt tuning | Li & Liang (2021) | $f_i' = f_{\boldsymbol{\theta}_i}([\boldsymbol{\phi}_i, \boldsymbol{x}])$ |
| | Parameter sharing | Ruder (2017) | $f_{\boldsymbol{\phi}_i}^t = f_{\boldsymbol{\phi}_i}^s \ \forall i \in \mathcal{G}$ |
| | Convolutional adapter | Rebuffi et al. (2017) | $f_i' = f_{\boldsymbol{\phi}_i}(f_{\boldsymbol{\theta}_i}(\boldsymbol{x}))$ where $f_{\boldsymbol{\phi}_i}(\boldsymbol{x}) = \boldsymbol{F} * \boldsymbol{x}$ |
| | Transformer adapter | Houlsby et al. (2019) | $f_i' = f_{\boldsymbol{\phi}_i}(f_{\boldsymbol{\theta}_i}(\boldsymbol{x}))$ where $f_{\boldsymbol{\phi}_i}(\boldsymbol{x}) = \boldsymbol{W}^d(\sigma(\boldsymbol{W}^u \boldsymbol{x}))$ |
| | Compacter | Mahabadi et al. (2021a) | $f_i' = f_{\boldsymbol{\phi}_i}(f_{\boldsymbol{\theta}_i}(\boldsymbol{x}))$ where $\begin{aligned} &f_{\boldsymbol{\phi}_i}(\boldsymbol{x}) = \boldsymbol{W}^d(\sigma(\boldsymbol{W}^u \boldsymbol{x})), \\ &\boldsymbol{W} = \sum_{j=1}^n \boldsymbol{A}_j \otimes \boldsymbol{B}_j \end{aligned}$ |
| | Parallel adapter | Rebuffi et al. (2018) | $f_i' = f_{\boldsymbol{\theta}_i}(\boldsymbol{x}) + f_{\boldsymbol{\phi}_i}(\boldsymbol{x})$ |
| | Rescaling | Bilen & Vedaldi (2017) | $f_i' = f_{\boldsymbol{\theta}_i}(\boldsymbol{x}) \odot \boldsymbol{\phi}$ |
| | Hypernetwork | Platanios et al. (2018) | $f_i' = (\boldsymbol{\alpha}\boldsymbol{W})\boldsymbol{x}$ |
| **Routing function** | Fixed routing | Hampshire & Waibel (1992) | $f_i' = \frac{1}{|K|} \sum_j f(\boldsymbol{\phi}_j) \mathbb{1}_j(K)$ |
| | Top-1 learned routing | Rosenbaum et al. (2018) | $f_i' = f(\boldsymbol{x}; \boldsymbol{\theta}_i, \boldsymbol{\phi}_j)$ where $j = \mathrm{argmax}[\boldsymbol{\alpha}]$ |
| | Top-$k$ learned routing | Goyal et al. (2021) | $f_i' = \mathrm{cat}_{j \in \mathrm{top}_k[\boldsymbol{\alpha}]} f(\boldsymbol{x}; \boldsymbol{\theta}_i, \boldsymbol{\phi}_j)$ |
| | Variable-size (threshold) | Rahaman et al. (2021) | $f_i' = \mathrm{cat}_{j \in M \,\mathrm{s.t.}\, \alpha_j > t} f(\boldsymbol{x}; \boldsymbol{\theta}_i, \boldsymbol{\phi}_j)$ |
| | Variable-size (soft partition) | Ponti et al. (2022) | $f_i' = \frac{1}{\sum \boldsymbol{\alpha}} \sum_{j \in M \,\mathrm{s.t.}\, \alpha_j = 1} f(\boldsymbol{x}; \boldsymbol{\theta}_i, \boldsymbol{\phi}_j)$ |
| | Mixture of experts | Jacobs et al. (1991b) | $f_i' = \sum_{j \in M} \alpha_j f(\boldsymbol{x}; \boldsymbol{\theta}_i, \boldsymbol{\phi}_j)$ |
| | Weighted top-$k$ routing | Shazeer et al. (2017) | $f_i' = \sum_{j \in \mathrm{top}_k[\boldsymbol{\alpha}]} \frac{\alpha_j}{\sum \boldsymbol{\alpha}} f(\boldsymbol{x}; \boldsymbol{\theta}_i, \boldsymbol{\phi}_j)$ |
| **Aggregation function** | Sparse weight addition | Ansell et al. (2022) | $f' = f_{\boldsymbol{\theta}_0 + \boldsymbol{\phi}_l + \boldsymbol{\phi}_t}$ |
| | Representation averaging | Ma et al. (2018) | $f_i' = \sum_j^{|M_i|} \alpha_j \boldsymbol{h}_j$ |
| | Input concatenation | Vu et al. (2022b) | $f_i' = f_{\boldsymbol{\theta}}([\boldsymbol{\phi}_t, \boldsymbol{\phi}_l, \boldsymbol{x}])$ |
| | Attention-based aggregation | Pfeiffer et al. (2021a) | $f_i' = \mathrm{Attn}(\boldsymbol{h}_{i-1}\boldsymbol{Q}_i, \boldsymbol{H}_i\boldsymbol{K}_i, \boldsymbol{H}_i\boldsymbol{V}_i)$ |
| | Sequential aggregation | Pfeiffer et al. (2020b) | $f_i' = f_{\boldsymbol{\phi}_t}(f_{\boldsymbol{\phi}_l}(f_{\boldsymbol{\theta}_0}(\boldsymbol{x})))$ |

Table 2: An overview of representative computation, routing, and aggregation functions. Each method is paired with a representative reference. In computation functions, skip connections are omitted for simplicity. Definitions: the model $f$, a model's sub-function $f_i$, model parameters $\boldsymbol{\theta}$, module parameters $\boldsymbol{\phi}$, parameters at initialisation $\boldsymbol{\theta}^0$, parameters after training $\boldsymbol{\theta}^\star$, binary mask $\boldsymbol{b} \in \{0, 1\}^{|\boldsymbol{\theta}|}$, random matrix $\mathbf{M}$, group $G$, input $\boldsymbol{x}$, a model's embedding layer $\mathrm{Emb}(\cdot)$, text prompt $\boldsymbol{p}$, retrieved context $\boldsymbol{c}$, filter bank $\boldsymbol{F}$, routing scores or task embedding $\boldsymbol{\alpha}$, routing function $r$, subset of modules $K$, module inventory $M$.

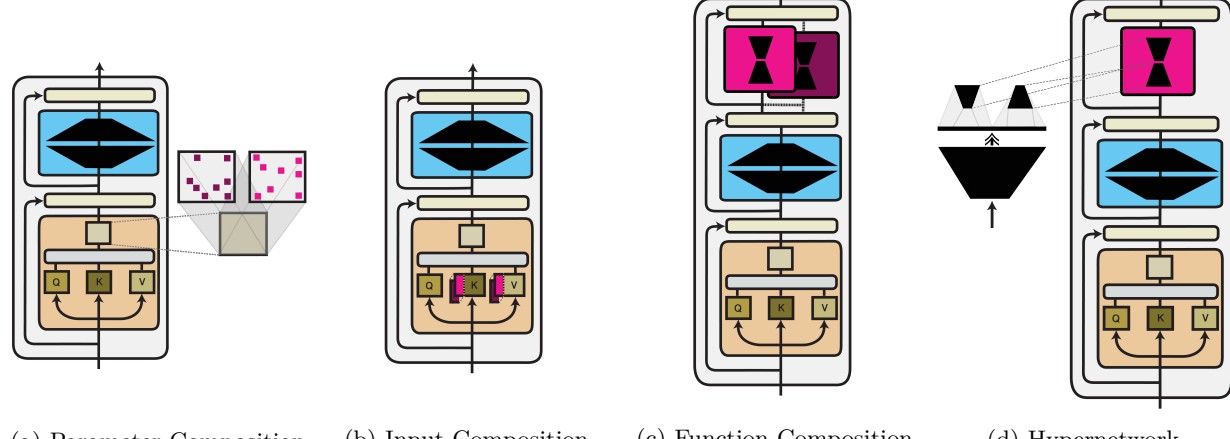

(a) Parameter Composition     (b) Input Composition     (c) Function Composition     (d) Hypernetwork

Figure 2: Different modular designs for Transformer architectures; best viewed in colour. Task-specific modular components are illustrated in magenta and purple, respectively. **(a) Parameter Composition** (§ 3.1): A sparse sub-network in the linear layer as part of multi-head-attention. **(b) Input Composition** (§ 3.2): Prefix-tuning (Li & Liang, 2021) extends the input by prepending embeddings to the key and value matrices in the Transformer layer. **(c) Function Composition** (§ 3.3): Task-specific bottleneck layers transforming the hidden representations are inserted in each layer (Houlsby et al., 2019). **(d) Hypernetwork** (§ 3.4): A small separate neural network generates modular parameters conditioned on metadata. We show its application to function composition but it is compatible with all computation functions.

As pruning generally leads to a loss in performance due to the change in network connections, the non-pruned weights are typically re-wound to their initialisation value and re-trained. In practice, rather than pruning all weights in a single run, iterative pruning is carried out over multiple stages (Han et al., 2015; Frankle & Carbin, 2019). The models pruned in this fashion often retain—if not surpass —the performance of the original dense model. The existence of a subnetwork with this property in any given randomly initialised model is known as the Lottery Ticket Hypothesis (LTH; Frankle & Carbin, 2019; Chen et al., 2020). These 'winning tickets' have also been shown to exist in RL and NLP (Yu et al., 2020), as well as in computer vision (Frankle et al., 2020). Subnetworks achieve above-random performance even when kept fixed at their random initialisation (Zhou et al., 2019; Wortsman et al., 2020; Zhao et al., 2020), so $f' = f_{\boldsymbol{\theta}_0 \odot \boldsymbol{b}}$. In this case, they are known as *supermasks*.

Winning tickets also occur in pre-trained models, such as language models (Chen et al., 2020; Prasanna et al., 2020). These often outperform tickets from randomly initialised models (Prasanna et al., 2020) and are less sensitive to specific hyper-parameter choices (Sun et al., 2020a). Magnitude pruning, which relies on zeroth-order information (the absolute value of a weight), is sub-optimal in this setting as fine-tuned weights typically stay close to their pre-trained values. Thus, magnitude pruning selects a similar set of weights for pruning regardless of the downstream task. Pruning based on first-order (gradient-based) information better captures the task-specific relevance of each weight (Molchanov et al., 2017). For instance, movement pruning (Sanh et al., 2020) learns the mask $\boldsymbol{b}$ jointly with the parameters $\boldsymbol{\theta}$. As the mask is a discrete binary variable, they rely on straight-through estimators (Bengio et al., 2013). Alternatively, $\boldsymbol{b}$ can be first learned as a real-valued mask and then binarised via a thresholding function (Mallya et al., 2018).

In addition to pruning, sparsification techniques can be employed for *adaptation*. In particular, a sparse module $\boldsymbol{\phi}$ can be merged with pre-trained parameters $\boldsymbol{\theta}$. For instance, in Sparse Fine-Tuning (SFT; Ansell et al., 2022) the LTH is re-purposed such that, instead of zeroing out weights with the lowest change in magnitude, they are simply frozen. Thus, only a subset of weights is fine-tuned.[7] The difference between these and the original pre-trained model results in a sparse module $\boldsymbol{\phi}$ where $\phi_i = 0$ if $b_i = 0$, which can be

---

[7]This is typically implemented by masking the gradient based on the binary mask $\boldsymbol{b} \odot \nabla_{\boldsymbol{\theta}} \mathcal{L}(f_{\boldsymbol{\theta}}, \mathcal{D})$ where $\mathcal{L}$ is a loss function and $\mathcal{D}$ is a dataset (Ansell et al., 2022).

| | Parameter efficiency | Training efficiency | Inference efficiency | Performance | Compositionality |
|---|---|---|---|---|---|
| Parameter composition | + | − | ++ | + | + |
| Input composition | ++ | − | − | − | + |
| Function composition | − | + | − | ++ | + |

Table 3: Comparison of computation functions along different dimensions. See the end of § 3.1 (parameter composition), § 3.2 (input composition), and § 3.3 (function composition) for further explanation. Compositionality is discussed in § 5.

plugged in and out of the model as $f'_{\boldsymbol{\theta}} = f_{\boldsymbol{\theta} \oplus \boldsymbol{\phi}}$. `Diff pruning` (Guo et al., 2021) instead obtains a sparse adapter by fine-tuning a dense difference vector $\boldsymbol{\phi}$ regularised to be sparse with a differentiable approximation to the $L_0$-norm penalty. Sung et al. (2021) induce a fixed sparse mask by selecting the top-$k$ weights ranked according to (a diagonal approximation of) their Fisher information. This second-order information reveals the impact of the change of a parameter on the model predictions. Thus,

$$\boldsymbol{b}_j = \begin{cases} 1 & \text{if } j \in \text{top-}k \ \frac{1}{n} \sum_{i=1}^n \mathbb{E}_{y \sim f_{\boldsymbol{\theta}^\star}(y|\boldsymbol{x}_i)} \left( \nabla_{\boldsymbol{\theta}} \log f_{\boldsymbol{\theta}^\star}(y \mid \boldsymbol{x}_i) \right)^2 \\ 0 & \text{otherwise} \end{cases} \tag{1}$$

Beyond the sparsification of individual weights, sparse model adaptation can also be *structured*. In this case, only a group of model sub-functions is fine-tuned, while the rest of the parameters remain frozen. The most common setting is for such a group to correspond to a subset of layers, e.g. the last one (Donahue et al., 2014). Groups can also relate to more fine-grained parts of the model. For instance, a group consisting of a model's bias parameters is a practical choice as this removes the need to store the model's intermediate activations (Cai et al., 2020; Ben Zaken et al., 2022). At the level of parameter tensors, some methods prune filters in CNNs (Anwar et al., 2017; Newell et al., 2019) whereas others prune attention heads in pre-trained Transformers (Voita et al., 2019; Michel et al., 2019). In structured diff pruning, members of a group are encouraged to share the same mask value (Guo et al., 2021).

**Low-Rank Modules**   Similar to sparsity, another efficient solution is for the module parameters $\boldsymbol{\phi}_i$ to lie in a low-dimensional subspace. Li et al. (2018) show that models can be optimised in a low-dimensional, randomly oriented subspace rather than the full parameter space. In this setting, the module parameters $\boldsymbol{\phi} \in \mathbb{R}^d$ are low-dimensional compared to the model parameters $\boldsymbol{\theta} \in \mathbb{R}^D$ and $d \ll D$. A random matrix $\mathbf{M} \in \mathbb{R}^{d \times D}$ can be used to project from $d$ to $D$: $f'_{\boldsymbol{\theta}} = f_{\boldsymbol{\theta} + \boldsymbol{\phi}\mathbf{M}}$. An efficient way to compute $\mathbf{M}$ is via the Fastfood transform (Le et al., 2014), which factorises $\mathbf{M}$ as random linear matrices. Specifically, $\mathbf{M} = \mathbf{HG\Pi HB}$ consists of a Hadamard matrix $\mathbf{H}$, a random diagonal matrix with independent standard normal entries $\mathbf{G}$, a random diagonal matrix with equal probability $\pm 1$ entries $\mathbf{B}$, and a random permutation matrix $\Pi$. Li et al. (2018) refer to the minimum $d$ that achieves within 90% of the full-parameter model performance as the intrinsic dimensionality of a given task. Aghajanyan et al. (2021) investigate the intrinsic dimensionality of various NLP tasks with different pre-trained models. They observe that it decreases during pre-training and that larger models have lower values.

However, storing the random matrices results in a substantial memory overhead and is slow to train (Mahabadi et al., 2021a). If the weight matrix $\boldsymbol{W} \in \mathbb{R}^{o \times i}$ is small enough, we can directly compose it into low-rank matrices $\boldsymbol{W} = \lambda \boldsymbol{B}\boldsymbol{A}$ where $\boldsymbol{A} \in \mathbb{R}^{k \times i}$ and $\boldsymbol{B} \in \mathbb{R}^{o \times k}$, where $i$ is the input dimensionality, $o$ is the output dimensionality, $k$ is the rank of the matrix, and $\lambda$ is a scaling hyper-parameter. To save space, the factorisation may be only applied to certain groups of parameters $G$. In LoRA (Hu et al., 2022), this group corresponds to the linear projections in the self-attention mechanisms of each Transformer layer: $f'_j = f_{\boldsymbol{\theta}_j + \text{vec}(\boldsymbol{B}_j \boldsymbol{A}_j)} \forall f'_j \in \mathcal{G}$.

Overall, parameter composition methods (both sparse and low-rank) are very parameter-efficient and often require updating less than 0.5% of a model's parameters (Guo et al., 2021). At inference time, they keep the model size constant or even reduce it, if the resulting model is sparse. This is compelling as it enables a plug-in replacement of the original with the modular model without any changes to the underlying architecture. Sparse modules, however, increase the time complexity of optimisation as they typically require multiple iterations of re-training. Finally, state-of-the-art parameter composition methods, e.g., LoRA (Hu et al., 2022) and SFT (Ansell et al., 2022) achieve strong performance in zero-shot and few-shot transfer.

## 3.2 Input Composition

Input composition methods augment a function's input $\boldsymbol{x}$ by concatenating it with a parameter vector $\boldsymbol{\phi}_i$: $f_i'(\boldsymbol{x}) = f_{\boldsymbol{\theta}_i}([\boldsymbol{\phi}_i, \boldsymbol{x}])$. The most common strategy is to augment the input fed to the model's first layer $f_1$.

**Prompting**   In a prompting setup with auto-regressive language models (Brown et al., 2020) or encoders (Schick & Schütze, 2021a;b), the input prompt $\boldsymbol{p}$ consists of (optional) instructions and (optional) in-context examples that have been converted to natural language. From a different perspective, the task-specific text prompt, when encoded using the model's embedding layer Emb($\cdot$), corresponds to modular parameters $\boldsymbol{\phi}$ that elicit the desired behaviour (Gao et al., 2021b; Liu et al., 2023): Emb($\boldsymbol{p}$) = $\boldsymbol{\phi}$. More elaborate prompts $\boldsymbol{p}$ such as rationales have led to improved few-shot reasoning performance (Wei et al., 2022c; Kojima et al., 2022; Shi et al., 2023). However, models are ostensibly sensitive to the formulation of the prompt as well as to the set and order of the (few-shot) examples (Zhao et al., 2021; Lu et al., 2022; Webson & Pavlick, 2022).

**Continuous Prompts**   Instead, a continuous prompt vector $\boldsymbol{\phi}$ can be learned directly (Lester et al., 2021; Liu et al., 2021b; Zhong et al., 2021; Hambardzumyan et al., 2021). However, if $\boldsymbol{\phi}$ is only concatenated with the first layer's input, the model has limited capacity to adapt to a specific task. As a result, such continuous (also called *soft*) prompts perform poorly at smaller model sizes and on some harder tasks (Mahabadi et al., 2021a; Liu et al., 2022c). To mitigate this, initialisation via multi-task learning has been proposed (Vu et al., 2022c). As an alternative, module vectors $\boldsymbol{\phi}_i$ can be learned *for each layer* of the model (Figure 2b; Li & Liang, 2021; Liu et al., 2022c). While this increases the number of parameters, it increases the modules' capacity to adapt to a given task. In practice, module parameters in the form of prefix vectors $\boldsymbol{\phi}_i = \boldsymbol{P}_k^i, \boldsymbol{P}_v^i \in \mathbb{R}^{l \times d}$ are prepended to the keys and values of every multi-head attention layer. Attention is defined as $f_i(\boldsymbol{x}) = \text{Attn}(\boldsymbol{x}\boldsymbol{W}_q^i, \boldsymbol{C}\boldsymbol{W}_k^i, \boldsymbol{C}\boldsymbol{W}_v^i)$ where $\boldsymbol{W}_q, \boldsymbol{W}_k, \boldsymbol{W}_v \in \mathbb{R}^{d \times d_h}$ are the projections that produce the queries, keys, and values, and $\boldsymbol{C} \in \mathbb{R}^{m \times d}$ is a sequence of context vectors. Multi-layer prompt tuning thus takes the following form:

$$f_i'(\boldsymbol{x}) = \text{Attn}(\boldsymbol{x}\boldsymbol{W}_q^i, [\boldsymbol{P}_k^i, \boldsymbol{C}\boldsymbol{W}_k^i], [\boldsymbol{P}_v^i, \boldsymbol{C}\boldsymbol{W}_v^i]). \tag{2}$$

**Retrieval Augmentation**   Beyond individual prompts, the input can be augmented with additional context $\boldsymbol{c}$ from a retrieval model. Retrieved documents are appended to the input and are used for conditioning the language model (Guu et al., 2020; Lewis et al., 2020a): Emb($[\boldsymbol{p}, \boldsymbol{c}]$) = $\boldsymbol{\phi}$.

In summary, input composition is exceptionally parameter-efficient as it only adds a very small number of parameters. However, these parameters extend a model's context window, which makes them less efficient during training and inference. Prompt tuning methods also require large models to achieve decent performance.

## 3.3 Function Composition

While parameter composition deals with individual weights and input composition methods act only on a function's input, function composition methods augment the model with new task-specific sub-functions (see Figure 2c): $f_i'(\boldsymbol{x}) = f_{\boldsymbol{\phi}_i} \circ f_{\boldsymbol{\theta}_i}(\boldsymbol{x}) = f_{\boldsymbol{\phi}_i}(f_{\boldsymbol{\theta}_i}(\boldsymbol{x}))$, where $\circ$ stands for function composition.

**Parameter Sharing**   Models in multi-task learning traditionally consist of shared layers $f_{\boldsymbol{\theta}}$ stacked under task-specific modules $f_{\boldsymbol{\phi}}$ (Ruder, 2017). Conversely, given models for tasks $t$ and $s$ expressed as a composition of functions $f_{\boldsymbol{\phi}_1}^t \circ \ldots \circ f_{\boldsymbol{\phi}_l}^t$ and $f_{\boldsymbol{\phi}_1}^s \circ \ldots \circ f_{\boldsymbol{\phi}_l}^s$, respectively, a multi-task architecture can also be obtained by tying sets of parameters between the models: $f_{\boldsymbol{\phi}_i}^t = f_{\boldsymbol{\phi}_i}^s \ \forall i \in \mathcal{G}$ where the group $\mathcal{G}$ contains the set of shared

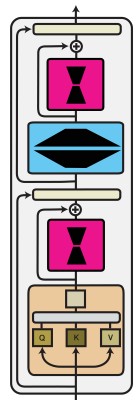

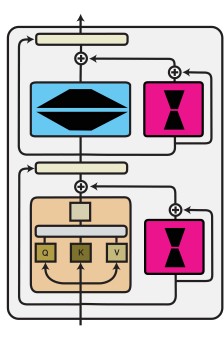

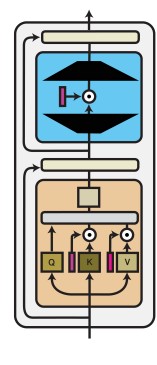

(a) Sequential
Bottleneck Adapter

(b) Parallel Bottleneck Adapter

(c) $(IA)^3$

Figure 3: Different approaches of function composition. **(a) Sequential Bottleneck Adapter:** The first adapter architecture proposed for transformers which consists of two bottleneck layers placed after the multi-head attention (MHA) and feed-forward (FF) layers (Houlsby et al., 2019). **(b) Parallel Bottleneck Adapter:** Bottleneck layers processed in parallel to the MHA and FF layers of the pre-trained transformer components (Rebuffi et al., 2018; Stickland & Murray, 2019; He et al., 2022a). **(c) $(IA)^3$:** Rescaling operations performed within the MHA and FF layers (Liu et al., 2022b).

layer indices.[8] Many multi-task neural architectures can be characterised in terms of their definition of $G$, which determines which modules are task-specific and which ones are shared. This is the case, for instance, of 'shared trunk' approaches in computer vision (Zhang et al., 2014; Ma et al., 2018) and approaches with supervision at different layers in NLP (Søgaard & Goldberg, 2016; Sanh et al., 2019; Liu et al., 2019).

Some approaches learn finer-grained interactions between pairs of modules. Misra et al. (2016) propose the cross-stitch unit, which linearly combines the inputs at every layer[9]: $(\widetilde{\boldsymbol{x}}^t, \widetilde{\boldsymbol{x}}^s) = \boldsymbol{W}[\boldsymbol{x}^t, \boldsymbol{x}^s]$ where

$$\begin{bmatrix} \widetilde{\boldsymbol{x}}^t_{ij} \\ \widetilde{\boldsymbol{x}}^s_{ij} \end{bmatrix} = \begin{bmatrix} \alpha^{tt} & \alpha^{ts} \\ \alpha^{st} & \alpha^{ss} \end{bmatrix} \begin{bmatrix} \boldsymbol{x}^t_{ij} \\ \boldsymbol{x}^s_{ij} \end{bmatrix}$$

and $\alpha \in \mathbb{R}$. Sluice networks (Ruder et al., 2019a) extend cross-stitch units to multiple modules per layer and additionally employ a soft selection of the skip connections from all layers at the output layer $l$:

$$\widetilde{\boldsymbol{x}}^{t\top} = \begin{bmatrix} \beta^t_1 \\ \cdots \\ \beta^t_l \end{bmatrix}^\top \begin{bmatrix} \boldsymbol{x}^{t\top}_1, & \dots, & \boldsymbol{x}^{t\top}_l \end{bmatrix}$$

and $\beta \in \mathbb{R}$. On the other hand, Gao et al. (2019) fuse features from multiple tasks through a 1x1 convolution. Bragman et al. (2019) employ variational inference to assign filters in a CNN to task-specific or shared roles.

Rather than learning which modules should be shared among which tasks, which is a combinatorially large problem, Lu et al. (2017) and Vandenhende et al. (2020) start with a fully shared model and then dynamically widen it during training, by cloning function $f_{\boldsymbol{\theta}_i}$ into new modules $f_{\boldsymbol{\phi}_{i,1}}, \dots, f_{\boldsymbol{\phi}_{i,k}}$ shared among a smaller subset of tasks, in top-down order across layers. More information on parameter-sharing strategies in multi-task learning can be found in relevant surveys (Ruder, 2017; Crawshaw, 2020).

---

[8]In this view, there is no clear differentiation between model parameters $\boldsymbol{\theta}$ and module parameters $\boldsymbol{\phi}$.

[9]We omit the layer index $n$ to simplify the presentation.

**Adapter Layers**  As an alternative to parameter sharing, a new task-specific learnable function $f_{\phi_i}$ can be composed with an (often frozen) shared function $f_{\theta_i}$. As the main purpose of such modules is adapting a pre-trained model to new tasks, they are also simply known as 'adapter layers'. We provide examples of different adapter layers in Figure 3.

The adapter's design and composition with the pre-trained model are often modality-specific. In computer vision, the adapter typically consists of a $1 \times 1$ convolution, i.e., $f_{\phi_i}(\boldsymbol{x}) = F * \boldsymbol{x}$ where $F$ is a bank of $\mathbf{1} \times 1$ filters and $*$ is the convolution operation (Rebuffi et al., 2017). The module is then inserted between the convolutional blocks of a pre-trained model, such as a ResNet (He et al., 2016). In NLP, a bottleneck architecture has become popular which consists of a down- and up-projection, coupled with an intermediate activation function $\sigma$: $f_{\phi_i}(\boldsymbol{x}) = \boldsymbol{W}^d(\sigma(\boldsymbol{W}^u \boldsymbol{x}))$ where $\boldsymbol{W}^d \in \mathbb{R}^{d_{\boldsymbol{x}} \times k}$ and $\boldsymbol{W}^U \in \mathbb{R}^{k \times d_{\boldsymbol{x}}}$, $d_{\boldsymbol{x}}$ is the dimensionality of the input (typically the hidden dimension), and $k$ is the bottleneck dimension. $\sigma$ is commonly a non-linearity such as a ReLU unit (Figure 3a; Houlsby et al., 2019; Pfeiffer et al., 2020b). In a Transformer model, adapters are placed both after the multi-head attention and the feed-forward layer (Houlsby et al., 2019), just after the multi-head attention (Bapna & Firat, 2019), or just after the feed-forward layer (Pfeiffer et al., 2020b).

Other variants for $\sigma$ such as the identity function, standard multi-head attention, and multi-head attention with shared projection matrices have also been explored (Stickland & Murray, 2019). Mahabadi et al. (2021a) propose Compacter, a hyper-complex, low-rank adapter that reparameterises $\boldsymbol{W}$ in the adapter as: $\boldsymbol{W} = \sum_{i=1}^{n} \boldsymbol{A}_i \otimes \boldsymbol{B}_i$ where $\boldsymbol{A}_i \in \mathbb{R}^{n \times n}$ is shared across layers ($n$ is a hyper-parameter), $\boldsymbol{B}_i \in \mathbb{R}^{\frac{k}{n} \times \frac{d}{n}}$ is parameterised as a low-rank matrix $\boldsymbol{B}_i = \boldsymbol{s}_i \boldsymbol{t}_i^{\top}$ and $\otimes$ is the Kronecker product.

Adapters can be routed sequentially or in parallel. Sequential adapters, are inserted between existing functions: $f_i'(\boldsymbol{x}) = f_{\phi_i}(f_{\theta_i}(\boldsymbol{x}))$ (Rebuffi et al., 2017; Houlsby et al., 2019). Parallel adapters are applied in parallel to a pretrained function: $f_i'(\boldsymbol{x}) = \boldsymbol{x} + f_{\theta_i}(\boldsymbol{x}) + f_{\phi_i}(\boldsymbol{x})$ (Figure 3b; Rebuffi et al., 2018; Stickland & Murray, 2019; He et al., 2022a). Moreover, adapters involve two residual connections: between the output of $f_{\theta_i}$ and the output of $f_{\phi_i}$, which is further added to $\boldsymbol{x}$ and normalised. Adapters have been shown to lead to increased sample efficiency, flatter minima, and more robustness to hyper-parameter choices compared to standard model fine-tuning (Mahabadi et al., 2021b; He et al., 2021; Han et al., 2021).

**Function Augmentation**  Adapters and more complex module designs can also be used to augment a base model with information and behaviour that it otherwise would not be able to access. This can be through adapter layers pre-trained on specific domains (Wang et al., 2021a) or other modalities (Alayrac et al., 2022). Modules can also be designed to attend over explicit key-value memory representations of entities and facts (Verga et al., 2021) and general domain knowledge (Cheng et al., 2023) to enable a model to perform certain types of operations such as arithmetic reasoning (Trask et al., 2018; Andor et al., 2019). More broadly, function composition enables the use of arbitrarily complex auxiliary modules. We highlight how function composition has been used to inject knowledge into models in §7.1.4. For an overview of how modules can be used to allow language models to use tools and to act, we direct the reader to Mialon et al. (2023).

**Rescaling**  The output representations can also be directly transformed via element-wise multiplication with a vector of learned parameters: $f_i'(\boldsymbol{x}) = f_{\theta_i}(\boldsymbol{x}) \odot \boldsymbol{\phi}$. Crucially, this is equivalent to stacking the original function $f_{\theta_i}$ with a linear transformation $\boldsymbol{W} = \boldsymbol{I}\boldsymbol{\phi}$. Such task-specific rescaling is typically applied to batch normalisation parameters in computer vision (Bilen & Vedaldi, 2017) and to layer normalisation parameters in NLP (Houlsby et al., 2019).

The adapter $(\text{IA})^3$ (Figure 3c; Liu et al., 2022b) multiplies learned vectors with the keys and values in self-attention blocks and the intermediate activations in position-wise feedforward networks in the Transformer. Rescaling activations favours dimensions that are important for a given task. Multiplication with a binary mask is a special case of rescaling that incorporates sparsity: Strezoski et al. (2019) multiply a task-specific random binary mask $\boldsymbol{b}$ with a function's input $\boldsymbol{x}$ at every layer.

Overall, standard function composition methods such as adapter layers typically require more parameters as the new function depends on a model's input size and hidden size. While they do not require storing the gradients of the frozen parameters, they increase the number of operations at training and inference time. State-of-the-art function composition methods match or outperform standard fine-tuning.

### 3.4 Hypernetworks

In the above-mentioned adapters, different modules $\boldsymbol{\phi}_1, \ldots, \boldsymbol{\phi}_{|M|}$ correspond to disjoint sets of parameters. However, the modules may benefit from sharing information. Rather than learning $\boldsymbol{\phi}_i$ directly, a (small) neural network $\boldsymbol{W}$, known as a hypernetwork, can generate the module parameters instead, conditioned on an embedding $\boldsymbol{\alpha}$ (Ha et al., 2017; Platanios et al., 2018). Thus, $\boldsymbol{\phi} = \boldsymbol{W}\boldsymbol{\alpha}$. As a result, the modules are 'entangled', which violates the strong definition of modularity that postulates that modules are autonomous (Goyal et al., 2021). In fact, in hypernetworks, computation and routing are inseparably intertwined. In fact, foreshadowing our discussion in § 4.2.4, the embedding $\boldsymbol{\alpha}$ can also be interpreted as unnormalised, learned routing scores for each task. In turn, the parameter generator weight would correspond to a set of modules stacked column-wise: $\boldsymbol{W} = [\boldsymbol{\phi}_1, \ldots, \boldsymbol{\phi}_{|M|}]$.

Hypernetworks can also be conditioned on inputs $\boldsymbol{x}$ (Figure 2d). For instance, in conditional batch normalisation (de Vries et al., 2017), rescaling parameters are generated based on a representation of the model input obtained via an LSTM. Feature-wise linear modulation (FiLM; Perez et al., 2018) generates an element-wise affine transformation that is applied to image features, conditioned on the linguistic input of the model, for text-and-vision tasks. In self-modulation for Generative Adversarial Networks (Chen et al., 2019), the affine transformation is applied to hidden representations of the generator conditioned on the noise sample. Bertinetto et al. (2016) conditions the parameter generator on individual examples, in order to perform one-shot learning.

Hypernetworks have been used to generate a diverse set of module parameters, including classifier heads (Ponti et al., 2021), continuous prompts (He et al., 2022c), and adapter layers (Üstün et al., 2020; Ansell et al., 2021; Mahabadi et al., 2021b), most commonly conditioned on task (Mahabadi et al., 2021b) or language embeddings (Üstün et al., 2020; Baziotis et al., 2022). Such task or language embeddings $\boldsymbol{\alpha}$ can themselves be learned directly from random initialisations or fixed as the typological features of a language (Üstün et al., 2020; Ansell et al., 2021). This is a strategy to integrate side (or metadata) information about the relationship among languages. Other examples of side information, such as the example label $y$, can be integrated into the hypernetwork input embedding via bi-linear interaction (Chen et al., 2019).

Nevertheless, even the smallest possible module generator network is a linear projection $\boldsymbol{W} \in \mathbb{R}^{d_\phi \times d_\alpha}$. To make the hypernetwork more parameter-efficient, it can be shared across layers by conditioning it on the module position in the neural architecture, in addition to the task index (Mahabadi et al., 2021b). In general, the hypernet can be conditioned on multiple (concatenated) embeddings: e.g., one corresponding to the task index and another to the language index. This allows the hypernetwork to generalise systematically to new task–language combinations at inference time. In particular, the hypernet can either generate a single module from all the embeddings (Ponti et al., 2021) or separate modules (Ansell et al., 2021; Üstün et al., 2022). In turn, the embedding combination chosen for any example is a form of hard routing (cf. § 4.2.2).

### 3.5 Unifying Parameter, Input, and Function Composition

While the above methods may seem different, they all covertly share a similar functional form. He et al. (2022a) cast LoRA (Hu et al., 2022), prefix tuning (Li & Liang, 2021), and bottleneck adapters (Houlsby et al., 2019), representative methods of the three composition functions, into the same framework. We extend their framework to cover parameter composition, input composition, and function composition in general. Specifically, all modular computation functions can be reduced to function composition: the output of the function $f_{\boldsymbol{\theta}_i}$ of a model is added to a new term that depends on a learned function $f_{\boldsymbol{\phi}}$: $f'_i(\boldsymbol{x}) = f_{\boldsymbol{\theta}}(\boldsymbol{x}) + f_{\boldsymbol{\phi}_i}(\boldsymbol{x})$.

For function composition methods, this form is the most natural. In the case of parallel adapters, for instance, $f'_i(\boldsymbol{x}) = f_{\boldsymbol{\theta}_i}(\boldsymbol{x}) + f_{\boldsymbol{\phi}_i}(\boldsymbol{x})$ where $f_{\boldsymbol{\theta}_i}(\boldsymbol{x})$ may be a multi-head attention module $f_{\boldsymbol{\theta}_i}(\boldsymbol{x}) = \text{MHA}(\boldsymbol{C}, \boldsymbol{x}) = [\text{head}_1, \ldots, \text{head}_h]\boldsymbol{W}_o$, with $\text{head}_j = \text{Attn}(\boldsymbol{x}\boldsymbol{W}_q^j, \boldsymbol{C}\boldsymbol{W}_k^j, \boldsymbol{C}\boldsymbol{W}_v^j)$, and $f_{\boldsymbol{\phi}_i}(\boldsymbol{x}) = \boldsymbol{W}^d(\sigma(\boldsymbol{W}^u\boldsymbol{x}))$. In this setting, $\boldsymbol{\theta}_i$ and $\boldsymbol{\phi}_i$ are independent and must only agree regarding the dimensionality of their inputs and outputs.

For parameter composition methods, which modify the parameters directly, the dimensionality of the module parameters $\boldsymbol{\phi}$ should match exactly the original parameters $\boldsymbol{\theta}_i$. For instance, if we apply the module to a linear projection, then they should consist of weight matrices $\boldsymbol{\theta}_i = \boldsymbol{W}_i \in \mathbb{R}^{d_x \times k}$ and $\boldsymbol{\phi}_i = \boldsymbol{V}_i \in \mathbb{R}^{d_x \times k}$,

respectively. Because of linearity:

$$f_i'(\boldsymbol{x}) = f_{\boldsymbol{\theta}_i \oplus \boldsymbol{\phi}}(\boldsymbol{x}) = f_{\boldsymbol{W}+\boldsymbol{V}}(\boldsymbol{x}) = (\boldsymbol{W} + \boldsymbol{V})\boldsymbol{x} = \boldsymbol{W}\boldsymbol{x} + \boldsymbol{V}\boldsymbol{x} = f_{\boldsymbol{\theta}_i}(\boldsymbol{x}) + f_{\boldsymbol{\phi}_i}(\boldsymbol{x})$$

For instance, in the case of LoRA (Hu et al., 2022), $\boldsymbol{V} = \lambda \boldsymbol{B}_i \boldsymbol{A}_i$. In the case of sparse adapters (Ansell et al., 2022), $\boldsymbol{V}$ is a sparse matrix.

For input composition methods, with the form $f_i'(\boldsymbol{x}) = f_{\boldsymbol{\theta}_i}([\boldsymbol{\phi}_i, \boldsymbol{x}])$, the equivalence is derived as follows. Prefix tuning (Li & Liang, 2021) generalises other continuous prompt methods by concatenating prefix vectors $\boldsymbol{\phi}_i = \boldsymbol{P}_k^i, \boldsymbol{P}_v^i \in \mathbb{R}^{l \times d}$ to the keys and values of self-attention. He et al. (2022a) show that prefix tuning can be expressed in the following way:

$$f_i'(\boldsymbol{x}) = \mathrm{Attn}(\boldsymbol{x}\boldsymbol{W}_q^i, [\boldsymbol{P}_k^i, \boldsymbol{C}\boldsymbol{W}_k^i], [\boldsymbol{P}_v^i, \boldsymbol{C}\boldsymbol{W}_v^i])$$
$$= (1 - \lambda(\boldsymbol{x})) f_{\boldsymbol{\theta}_i}(\boldsymbol{x}) + \lambda(\boldsymbol{x}) \, \mathrm{softmax}(\boldsymbol{x}\boldsymbol{W}_q \boldsymbol{P}_k^\top) \boldsymbol{P}_v$$

where $\lambda(\boldsymbol{x})$ is a scalar that represents the sum of normalised attention weights on the prefixes and $f_{\boldsymbol{\theta}_i}(\boldsymbol{x})$ is the attention module in a Transformer. If we set, $f_{\boldsymbol{\phi}_i}(\boldsymbol{x}) = \mathrm{softmax}(\boldsymbol{x}\boldsymbol{W}_q \boldsymbol{P}_k^\top) \boldsymbol{P}_v$, then we obtain a function composition $(1 - \lambda(\boldsymbol{x})) f_{\boldsymbol{\theta}_i}(\boldsymbol{x}) + \lambda(\boldsymbol{x}) f_{\boldsymbol{\phi}_i}(\boldsymbol{x})$ that incorporates a weighted addition. For function and parameter composition, in contrast, the sum is unweighted.

Overall, despite their conceptual differences, most modular approaches are similar in their functional form and can be expressed as function composition. In practice, the way different methods are realised, however, leads to different trade-offs, which we illustrate in Table 3. Recent empirical studies (Mahabadi et al., 2021a; He et al., 2022a; Liu et al., 2022b) provide further evidence for the strengths and weaknesses of different methods. For instance, prompt tuning (Vu et al., 2022a) underperforms other methods due to limited capacity while intrinsic dimensionality (Aghajanyan et al., 2021) uses a very small number of parameters but leads to a large memory footprint and poor performance. Fine-tuning only biases (Ben Zaken et al., 2022) has a small memory footprint but achieves lower performance. Finally, function composition methods such as adapter layers (Pfeiffer et al., 2021a) and compacter layers (Mahabadi et al., 2021a), achieve the best performance, but add more parameters. (IA)$^3$ (Liu et al., 2022b) mitigates this by composing a lightweight linear diagonal weight. Modular deep learning architectures, however, have many other differences beyond their choice of computation function. In the following sections, we discuss the routing, aggregation, and training settings for the modules presented so far.

- Computation functions may consist of any neural module. Modules may modify the original **parameters**, be concatenated to the **input**, or composed with the original **function**.
- **Parameter composition** methods utilise sparsity or low-rank constraints. They are very parameter-efficient and efficient at inference time and show strong performance.
- **Input composition** methods concatenate a function's input with a parameter vector via prompting, continuous prompts, and retrieval augmentation. They are extremely parameter-efficient but inefficient during training and inference and require large models.
- **Function composition** methods augment a model with arbitrary functions via parameter sharing, adapters, or rescaling. They require more parameters but often achieve the best performance.
- Rather than learning module parameters directly, **hypernetworks** can be used to generate module parameters, which enables sharing of information and conditioning on auxiliary information.

## 4 Routing Function

In the previous section, we described how to compose a sub-function $f_i$ with shared weights $\boldsymbol{\theta}$ with a single module function with weights $\boldsymbol{\phi}$. However, in a modular neural architecture, *multiple* modules are available from an inventory $M = f_{\boldsymbol{\phi}_1}, \ldots, f_{\boldsymbol{\phi}_{|M|}}$. A decision-making process is required to determine which modules are active, conditioned on the model input or auxiliary metadata. This process is implemented through a routing function $r(\cdot)$ that assigns a score $\alpha_i$ to each module from the inventory $M$. These scores determine

which subset of modules is active, i.e. contributes to the computation. We provide an overview of different routing methods in Figure 4.

When metadata such as expert knowledge about sub-tasks (or skills) involved in a task is available, $r(\cdot)$ can be designed as a *fixed* function, that is, each routing decision can be made *a priori* (Figure 4a). For instance, when using a language model to generate dialogue in Swahili, a task module for dialogue generation and a language module for Swahili can be selected. When no such prior information is available—for instance when modelling heterogeneous unlabelled data—routing of a given example needs to be *learned* (Figures 4b-4c). In this case, the routing function can be conditioned on the current example $\mathbf{x}$.[10]

Unfortunately, learned routing is crucially under-constrained, as multiple possible ways of decomposing tasks into sub-tasks are reasonable (Jacobs et al., 1991a). In addition, it presents a series of unique challenges (see § 4.2.1). In an empirical study on synthetic data, Mittal et al. (2022) found that learned routing is sub-optimal compared to fixed routing, as it tends to under-utilise modules and to specialise them to a lesser degree. This behaviour is exacerbated as the number of tasks in the data grows. In real-world applications, Muqeeth et al. (2022) report similar results; however, Ponti et al. (2022) find that learned routing may surpass expert module selection even in settings where tasks are procedurally constructed to require certain skills, such as instruction following in simulated environments.

Learning-to-route can roughly be split into *hard* routing and *soft* routing (Rosenbaum et al., 2019). *Hard* routing methods learn a binary selection of modules, similarly to the fixed routing scheme, where only a subset of modules is selected for each decision-making step (Figure 4b). Inference for hard routing systems typically builds on score function estimators (Williams, 1988; 1992) or stochastic re-parameterisation (Jang et al., 2017). On the other hand, *soft* routing methods learn a probability distribution over modules (Figure 4c; Jacobs et al., 1991a). While soft selection is more easily amenable to end-to-end learning via gradient descent, hard selection may lead to a sparse architectural design, owing to the fact that inactive modules are not part of the forward and backward computation graph. This reduces time complexity while augmenting the model capacity (Bengio et al., 2013).

While not the central focus of this paper, routing algorithms have recently garnered significant attention, due to their efficiency implications. There exists an intricate interplay between routing techniques and sparsity within modular models: When a modular architecture exhibits sparsity, signifying that only a select few modules are active during inference, a notable reduction in computational complexity during inference can be achieved (Fedus et al., 2022).[11] Finally, Shen et al. (2023a); Jang et al. (2023) showcase how sparse models, when combined with instruction tuning techniques, can yield substantial gains in performance and efficiency over dense models, potentially reshaping the landscape of large language model design.

## 4.1 Fixed Routing

Making the routing decision *a priori*—i.e. when we utilise metadata (e.g. task identity $t$) to make the discrete routing decisions *before* training—is referred to as fixed routing (Figure 4a). Here the routing function $r(\cdot)$ simplifies to a selection of a subset of modules $K \subseteq M$ for the examples with certain metadata:

$$r(\phi_i) = \begin{cases} 1 & \text{if } i \in K \\ 0 & \text{otherwise} \end{cases} \tag{3}$$

This function defines a binary matrix $A \in \{0,1\}^{|\mathcal{T}| \times |M|}$, where the number of rows corresponds to possible tasks and the number of columns corresponds to the size of the module inventory.

One simple example of fixed routing in multi-task learning is when all parameters, except the final classification layer, are shared among all tasks (Ruder, 2017). Independently from the task identity, the examples are passed through the same network until after the penultimate layer. The penultimate layer's representations

---

[10]Alternative non-parametric routing strategies include random routing (Zuo et al., 2022; Wang et al., 2022) or routing based on hash functions (Roller et al., 2021).

[11]In §4.2.3 "Token-Level Routing" we provide a brief overview over recent works that focus on the efficiency aspect. However, it is important to note, that the focus of this paper is centred around modularity and not efficiency.

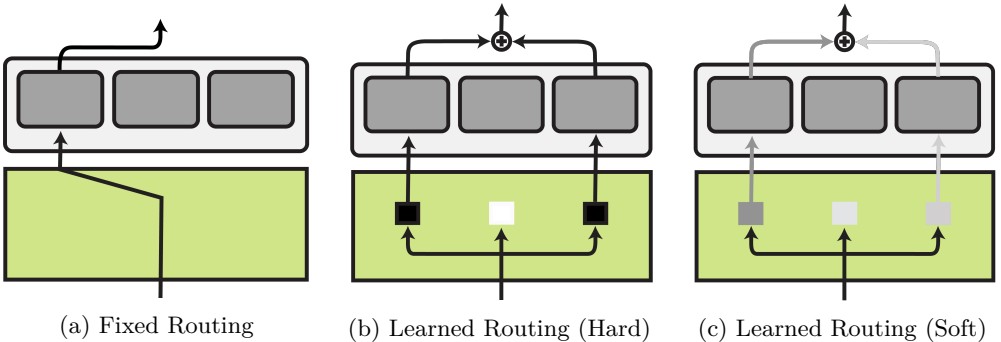

|  |  |  |
|:---:|:---:|:---:|
| (a) Fixed Routing | (b) Learned Routing (Hard) | (c) Learned Routing (Soft) |

Figure 4: Different routing methods. (a) **Fixed Routing:** Examples are passed to modules based on a pre-defined logic, known a priori. (b) **Hard Learned Routing:** Learned hard selection modules. (c) **Soft Learned Routing:** Soft selection and weighting of modules.

are then *routed* to their respective final classification layer according to the task identity. This boils down to setting $|K| = 1$, with the additional constraint that tasks cannot share modules, which results in the allocation matrix being an identity matrix, $A = I$.

While not immediately apparent, methods that adapt pre-trained models towards individual tasks (Rebuffi et al., 2017; 2018; Houlsby et al., 2019; Bapna & Firat, 2019; Li & Liang, 2021; Liu et al., 2022b; Hu et al., 2022; Ansell et al., 2022; Ben Zaken et al., 2022, *inter alia*)–as discussed in § 3–deterministically route representations through the newly introduced module $f_\phi$. Given that the pre-trained weights are frozen and modules trained on different tasks can be added or removed, the components become modular even if they are developed asynchronously and independently of each other (Pfeiffer et al., 2021a). In a sense, community-based hubs of pre-trained adapters such as AdapterHub (Pfeiffer et al., 2020a) can be considered as ever-evolving multi-task models, the development of whose components has been distributed throughout the community.[12] Moreover, since newly introduced weights are encapsulated between frozen (shared) weights, adapted representations of intermediate layers are implicitly aligned as they are passed as input to the same frozen components.

Hampshire & Waibel (1992) were possibly among the first to train independent experts for a series of sub-tasks known *a priori*. In this case, the (fixed-size) subset of experts $K$ associated with each task $t$ is assumed as given, resulting in the rows of $A$ being $k$-way vectors. In cross-lingual transfer, any problem can be decomposed into a task and language variety. Fixed routing can select separate language and task components, and facilitate generalisation to new, unobserved combinations of tasks and languages at inference time (Pfeiffer et al., 2020b; Ponti et al., 2021; Üstün et al., 2022). In this case, $|K| = 2$. Similarly, in reinforcement learning, Heess et al. (2016) and Devin et al. (2017) design a modular policy that is composed of a robot-specific module and a task-specific module, which are instantiated as separate neural networks. Composing these modules enables generalisation to unseen robot–task combinations.

Beyond task identity, routing can be performed based on other metadata such as language, domain, or modality information. Pfeiffer et al. (2022b) add adapters for each language to a multilingual language model during pre-training on unlabelled text. Fan et al. (2021) route deterministically for multilingual machine translation according to the language family: as a consequence, all languages in a family share the same expert. In a similar vein, Gururangan et al. (2022) add domain-specific adapters to language models, deterministically routing based on the text source domain. This concept was further extended by Li et al. (2022b), who proposed the branch–train–merge method: copies of the same model are trained on different domains and then averaged. Finally, modality can also inform fixed routing, such as in vision-and-language models (Pfeiffer et al., 2022a). This allows for adapting the encoders of different modality streams.

---

[12]Alternatively, combining entire models stored in model repositories via distillation (Khanuja et al., 2021) or averaging (Matena & Raffel, 2021) can also help avoid negative interference (Don-Yehiya et al., 2022); however, this is usually less efficient and subject to limitations such as those discussed later in § 5.

### 4.2 Learned Routing

When the routing function $r(\cdot)$ is not known in advance, it can be implemented as a learnable neural network with parameters $\boldsymbol{\rho}$. In input, it receives the example representation $\boldsymbol{x}$ or metadata such as the task $t$. In output, it returns routing scores $\boldsymbol{\alpha}$. Usually, $r_{\boldsymbol{\rho}}$ is a linear projection or a Multi-Layer Perceptron. While the former represents a less expressive family of functions, the latter may collapse into ignoring the input features. Note that learning the routing function also implies that the specialisation of each module is unknown. Thus, modules are not trained on different sets of examples; rather, they are all trained jointly with the routing function.

#### 4.2.1 Challenges of Learned Routing

Learned routing introduces a number of challenges, including *training instability*, *module collapse* (Kirsch et al., 2018), and *overfitting*. These were first systematically described by Rosenbaum et al. (2019), and we follow a similar taxonomy. In general, they identify two root causes for all these challenges: first, the need to balance between exploration and exploitation (Sutton, 1986). More specifically, routing must find the optimal trade-off between allocating information to the most suitable modules versus under-explored modules. Second, routing must share modules across examples or tasks in such a way as to reap the benefits of positive transfer while avoiding negative interference. We elaborate on the individual challenges below.

**Training Instability** emerges especially in the early phases of training; at this point, modules are randomly initialised and have no clear functional specialisation. Thus, the router cannot make any principled decision in selecting modules. On the other hand, modules do not start specialising until they are consistently routed to different subsets of tasks or examples.

Curriculum learning can mitigate this challenge to some extent (Chang et al., 2019), as simpler tasks require simpler sets of skills. However, this assumes that information about task complexity is available and that the data can be ordered accordingly. As an alternative, the router parameters can be trained with a different learning rate than the module parameters, either lower (Rosenbaum et al., 2018) or higher (Ponti et al., 2022). These create two different dynamics: either the necessary skills for a task are determined after specialisation, or the relationship among tasks is figured out first and modules are updated accordingly.

**Module Collapse** describes scenarios where only a small number of modules (in the extreme case, one) from the available inventory are selected. This leaves the remaining modules untrained and negatively impacts their overall diversity. Often, this results from excessively favouring exploitation over exploration, which leads to sub-optimal results. To amend this, Ahn et al. (2019) use $\epsilon$-greedy routing for initial exploration of all modules and afterwards switch to learned routing. Other strategies to avoid module collapse include auxiliary losses for load balancing (Shazeer et al., 2017; Fedus et al., 2021) and intrinsic rewards that encourage diversity in module selection (Cases et al., 2019). The choice of information that conditions the router also plays an important role: metadata, e.g. text genre (Cases et al., 2019) or task identity (Kudugunta et al., 2021), make routing more robust than individual examples. The diversity of training tasks also facilitates diversity in routing selections (Chang et al., 2019; Caccia et al., 2022). Dua et al. (2022) warms up the sampling temperature over training, in order to over-sample domains with fewer examples in unbalanced distributions.

**Overfitting** to noise is a risk faced by deep modular networks due to their ability to model subsets of examples independently (Rosenbaum et al., 2019). For instance, routing at the token level was shown to lead to performance drops in out-of-domain generalisation for MoEs (Artetxe et al., 2022). For a similar reason, gains in pre-training do not always translate into gains in fine-tuning for MoEs (Fedus et al., 2021). Increased robustness can be achieved by routing conditioned on metadata if available (Chang et al., 2019; Cases et al., 2019; Kudugunta et al., 2021). In addition, strategies that favour the combinatorial behaviour of modules yield superior generalisation (Chang et al., 2019; Ponti et al., 2022).

#### 4.2.2 Hard Learned Routing

A model may learn how to select modules through *hard* routing. This implies that the choice of whether a module is active or excluded from the computation graph is binary. Discrete decisions are not amenable

to be learned through vanilla gradient descent: since small perturbations of parameters do not affect the selection of modules, the gradient of the loss with respect to the routing parameters is zero. Thus, various methods, including reinforcement learning, evolutionary algorithms, and stochastic re-parameterisation, have been proposed for inference. These are discussed separately below.

On the other hand, hard routing is more efficient than soft routing in terms of time and space complexity. In addition, binary selection implies that parameter updates are localised to a subset of modules. This reflects the intuition that the shifts in distribution of the variables in an environment are similarly local (Parascandolo et al., 2018; Goyal et al., 2021). Since the inactive module parameters are not affected, they remain invariant with respect to the distribution shift. On top of this, this type of routing may result in variable-size sets of active modules. This allocates model capacity according to task complexity, which follows the principle of conditional computation (Bengio et al., 2015). In fact, it is fair to assume that the skills required for complex tasks are a superset of those of simpler tasks. For instance, dialogue modelling requires (among others) intent detection, slot filling, and conditional response generation.

**Reinforcement Learning** In Routing Networks (Rosenbaum et al., 2018), Modular Networks (Kirsch et al., 2018), and the Compositional Recursive Learner (CRL; Chang et al., 2019), a router network is trained through reinforcement learning. Specifically, Routing Networks rely on multi-agent RL (MARL), Modular Networks rely on the score function estimator (REINFORCE), whereas the CRL relies on Proximal Policy Optimisation (PPO). Commonly, this family of methods alternate between a score function estimator for the routing parameters $\boldsymbol{\rho}$ and SGD for module parameters $\{\boldsymbol{\phi}_1, \ldots, \boldsymbol{\phi}_{|M|}\}$. For a vanilla score function estimator, where routing is conditioned on the input example and $m \in M$, the update takes the form:

$$\nabla_{\boldsymbol{\rho}} \mathbb{E}_{\boldsymbol{x},\boldsymbol{y}}\, p(\boldsymbol{y} \mid \boldsymbol{x}, \boldsymbol{\theta}, \boldsymbol{\phi}_1, \ldots, \boldsymbol{\phi}_{|M|}, \boldsymbol{\rho}) \approx \frac{1}{n} \sum_{i=0}^{n} \left[ p(\boldsymbol{y}_i \mid \boldsymbol{x}_i, \boldsymbol{\theta}, \boldsymbol{\phi}_m)\, \nabla_{\boldsymbol{\rho}} \log p(m \mid \boldsymbol{x}_i) \right] \quad (4)$$

Under this lens, routing becomes a policy $\pi(m \mid \boldsymbol{x})$. If applied layer-wise, each hidden representation at a given layer $1 \geq t \leq l$ constitutes a state $\boldsymbol{h}_t \in \mathcal{H}$. The routing policy determines the action, i.e. the selection of a module index $m$. In particular, this assumes that the inventory $M$ is shared across layers.[13] In turn, applying the transformation of the corresponding module on the input is equivalent to a transition function $\pi : \mathcal{H} \to \mathcal{H}$, which returns the next layer's hidden state $\boldsymbol{h}_{t+1}$. The loss function at the top layer corresponds to a (delayed) negative reward, i.e. $\mathcal{L}(\cdot) = -\mathcal{R}$.[14] Crucially, in this setting the transition functions are non-stationary, as the module parameters are amenable to change. Because modules are applied sequentially based on the policy, the number of steps of computation in the model can vary when a special halting action is available.

**Evolutionary Algorithms** Alternatively, routing can be learned via a genetic algorithm. In PathNet (Fernando et al., 2017), the loss function indicates the fitness of a configuration of active modules $K \subseteq M$. For each task, two configurations are selected at random and trained until a stopping criterion is met. The one incurring the lower loss on a validation set overwrites the other. This copy, in turn, receives a random mutation, and then the procedure is repeated. In $\mu$Net (Gesmundo & Dean, 2022a;b), mutations involve cloning, insertion, and removal of layers. The fitness criteria include not only performance but also parameter efficiency. This approach has been extended to a multi-task setting where multiple agents update different modules asynchronously (Gesmundo, 2022). However, as is common for evolutionary algorithms, this search is brute-force and thus highly inefficient.

**Stochastic Re-parametrisation** Hard routing can also be performed via a continuous relaxation of the discrete latent variable $\boldsymbol{\alpha}$ determining the module allocation. Several stochastic re-parameterisations such as Gumbel-Softmax (Jang et al., 2017) or the Concrete distribution (Maddison et al., 2017) have been proposed for this purpose. Compared to the score function estimator, stochastic re-parameterisations are biased but have lower variance. Moreover, they are differentiable, which makes a hard router trainable in an end-to-end

---

[13]This encourages module re-usage at different layers.
[14]Intrinsic rewards can be added, for instance favouring diversity in the module selection across time steps (Rosenbaum et al., 2018).

fashion. For instance, AdaShare (Sun et al., 2020b) uses Gumbel-Sigmoid to learn a binary vector for each task that indicates whether a model layer should be included in the forward pass or skipped entirely. This may be interpreted as choosing between a parameterised module and an identity function at each layer.

Stochastic re-parameterisation also allows for selecting module subsets of *varying sizes* for each layer. In Neural Interpreters (Rahaman et al., 2021), this is based on a threshold. Each module is associated with a 'signature vector'. The dot product between this vector and the output of an unnormalised routing function ('type inference') conditioned on a token determines a score. If this surpasses a certain threshold, then the module is allowed to access the given token. As an alternative, variable-size module routing can be achieved by learning a soft clustering (a.k.a. soft partition) of modules (Ponti et al., 2022; Caccia et al., 2022). Thus, each entry $\alpha_{ij}$, which represents the routing of the $j$-th module to the $i$-th task, is constructed as follows:

$$\alpha_{i,j} = \text{sigmoid}\left[\log\frac{\text{sigmoid}(\hat{\alpha}_{i,j})\,u}{(1-\text{sigmoid}(\hat{\alpha}_{i,j}))\,(1-u)}^{1/\tau}\right] \qquad u \sim \text{Uniform}(0,1). \tag{5}$$

where $\hat{\alpha}_{ij}$ represents the unnormalised routing score. This latent variable also admits priors such as the Indian Buffet Process (Griffiths & Ghahramani, 2011) to encourage both diversification and sharing of module subsets across tasks (Ponti et al., 2022). Caccia et al. (2022) extend this framework to multi-head routing, where different modules can be allocated to contiguous subsets of dimensions of the layer's input and output. While this just requires as many copies of $\alpha$ as the number of subsets of dimension, it provides higher expressivity to the routing function.

**Top-$k$ Selection**   Finally, hard selection can rely on top-$k$ selection from (possibly unnormalised) scores $\boldsymbol{\alpha}$ over modules. In the case of Independent Causal Mechanisms (Parascandolo et al., 2018), $\boldsymbol{\alpha}$ is given by a discriminator that scores the outputs of a generator, and $k = 1$. In the case of Recurrent Independent Mechanisms (Goyal et al., 2021), the scores are derived from attention between modules and the input, and $k > 1$. These methods are grounded on the assumption that the competition among modules to be activated facilitates their specialisation (see § 8.3 for more details).

### 4.2.3   Soft Learned Routing

**Mixture of Experts**   To sidestep discrete selections of modules, several works propose *soft* routing methods, where all modules are selected and aggregated according to a *weighted combination*, i.e. a mixture of experts (MoE; Jacobs et al., 1991b; Jordan & Jacobs, 1994).[15] Here, the router learns a probability distribution over the available modules, i.e. $p(M) = r_{\boldsymbol{\rho}}(\cdot)$. Hence, routing and aggregation take place as:

$$f_i'(\boldsymbol{x}) = \sum_{\boldsymbol{\phi}_j \in M} r(\boldsymbol{\phi}_j)\,f(\boldsymbol{x};\boldsymbol{\theta}_i,\boldsymbol{\phi}_j) \tag{6}$$

In contrast to the discrete selection of hard routing methods, this setup is easily trained end-to-end via gradient descent. A number of works (Eigen et al., 2014; Meyerson & Miikkulainen, 2018; Wortsman et al., 2020, *inter alia*) train a continuous weighting (i.e. a mixture) of all modules; however, this limits the degree of modularity as parameter updates are not local; instead, they always affect all modules. Additionally, activating *all* modules for each example significantly increases the computational cost for each forward and backward pass through the network. To circumvent this, Shazeer et al. (2017) and Lepikhin et al. (2021) only route to the top-$k$ of $|M|$ modules, where $1 < k < |M|$. The output representations of the $k$ active modules are averaged according to the respective routing weights, whose sum is re-normalised to 1. Thus, top-$k$ MoEs stand between hard routing, as only a subset of modules is active, and soft routing, as their average is weighted by the routing scores. In practice, a layer performs the following computation:

$$f_i'(\boldsymbol{x}) = \sum_{\boldsymbol{\phi}_j \in \text{top}_k[r(\boldsymbol{\phi})]} \frac{r(\boldsymbol{\phi}_j)}{\sum_1^k r(\boldsymbol{\phi})}\,f(\boldsymbol{x};\boldsymbol{\theta}_i,\boldsymbol{\phi}_j) \tag{7}$$

---

[15]In the following sections we use the term "expert" and "module" interchangeably to reflect common practice in the body of research on MoEs.

Fedus et al. (2021) and Clark et al. (2022) demonstrate that even top-1 routing can achieve competitive results for language modelling.

**Token-Level Routing**   MoEs have recently undergone a revival as part of the efforts to scale Transformers. In particular, MoE Transformers route to a subset of Feed-Forward Network (FFN) modules per layer instead of a single FFN. The focus of these works is on computationally efficient training of very large models. This is achieved by splitting the input tokens across different (hardware) accelerators. The MoE routing algorithm is therefore required to (ideally) uniformly distribute the tokens of all the examples in an input batch across all accelerators, i.e. to *load balance* computation across "experts". The dominating routing strategy is for each *token* to choose the top-*k experts* (Shazeer et al., 2017; Lepikhin et al., 2021; Fedus et al., 2021; Clark et al., 2022; Yang et al., 2021; Dua et al., 2022; Hazimeh et al., 2021; Rajbhandari et al., 2022; Riquelme et al., 2021; Du et al., 2022; Zoph et al., 2022). Alternative approaches let each *expert* choose the top-*k tokens* (You et al., 2022; Zhou et al., 2022b) or *globally* determine the best routing path (Lewis et al., 2021).[16]

However, since routing is conditioned on the token level, and the *load balancing* restriction limits the system from routing an entire example to a single module, the system potentially has to relearn similar concepts in multiple modules. Hence, load balancing hinders the router from selecting the single best module for longer (e.g., repetitive) sequences. This is investigated further by Lewis et al. (2021), who find that sparse models route syntactically and semantically similar words (in contrast to sentences or phrases) to the same modules. This sheds light on the limited expressiveness of modules which are learned on the token-level. Since scaling is the main focus of these works, their goals are orthogonal to modular approaches centred on parameter efficiency, transfer–interference trade-offs, and combinatorial generalisation.

**Example-Level Routing**   Nevertheless, one could imagine obtaining the best of both worlds by hybridising sparse MoE Transformers models with deterministic or learned routing strategies from § 4.1 and § 4.2.2. Instead of routing each individual token separately, all tokens of a single example can be routed to the same experts. Kudugunta et al. (2021) experiment with two versions of example-level routing for machine translation: In *sentence-level* routing, they average pool over the token embeddings, and condition the router on the resulting representation. In *task-level* routing, a task embedding is trained, based on which the router learns the distribution over modules. In a similar vein, Gupta et al. (2022) and Xi et al. (2022) implement *task-level* routing across modular experts to improve the amount of knowledge sharing during multi-task learning in NLP and computer vision, respectively.

Since task identity (or other metadata) is not always given, especially in continual learning, it can be inferred through an auxiliary model. Van de Ven & Tolias (2019) refer to this scenario as '*class-incremental learning*'. For instance, the current task can be identified based on the lowest predictive uncertainty or an auxiliary task classifier (von Oswald et al., 2020). In these cases, routing can depend on the predicted task identity.

**Mitigating Module Collapse**   To address the challenge of module collapse, which was previously discussed in §4.2.1, several strategies have been introduced to enhance the effectiveness of models in utilizing the available experts' capacity. One such approach, presented by Shen et al. (2023b), introduces a novel loss function centered on Mutual Information. This loss aims to maximize the mutual information between the input and the target module, effectively mitigating module collapse issues. Another innovative solution, put forward by Chi et al. (2022), involves the modification of the routing algorithm. This modification incorporates techniques like dimension reduction, L2 normalization, and adjustment of gating temperature, all designed to address the challenges associated with module collapse. Puigcerver et al. (2023) employ a fully differentiable soft assignment mechanism by applying weighted combinations of representations to each module, allowing for enhanced model capacity without significantly increasing inference costs. Muqeeth et al. (2023) tackle module collapse by employing a weighted average-based merging approach of the module's parameters.

---

[16]For more details on load balancing methods we refer to Fedus et al. (2022), Chapter 4.

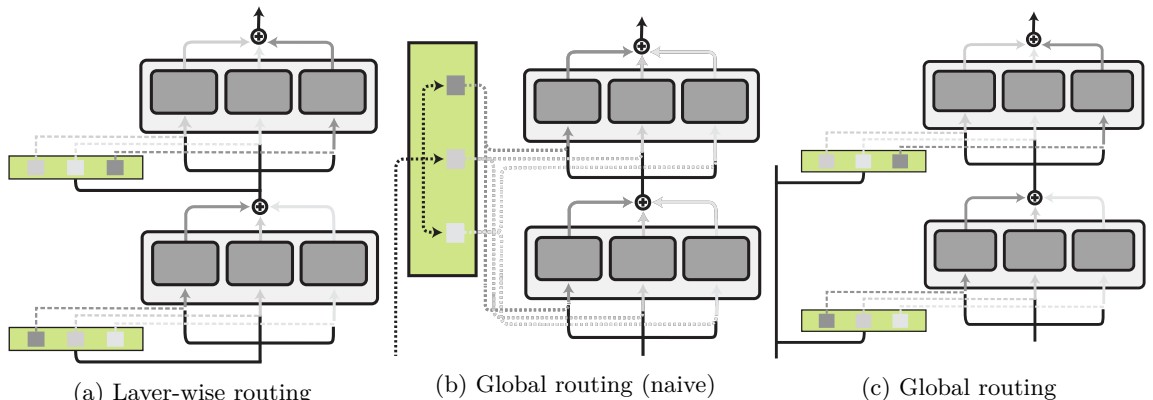

(a) Layer-wise routing      (b) Global routing (naive)      (c) Global routing

Figure 5: Different routing levels. (a) **Layer-wise Routing:** The indices are chosen based on the input to the current layer. (b) **Naive Global Routing:** The same indices of modules are chosen for all the layers of the model. (c) **Global Routing:** The configuration (possibly different for each layer) is chosen globally.

### 4.2.4  Hypernetworks

In addition to hard and soft routing, hypernetworks (Ha et al., 2017), as introduced in § 3.4, can be considered a third kind of routing, with unnormalised routing scores. More formally, the parameters $\theta_t \in \mathbb{R}^d$ for a task $t$ can be generated by a linear function $\Phi\alpha_t$. The task embedding $\alpha_t \in \mathbb{R}^{|M|}$ can be interpreted as the output of a task-level routing function with unnormalised scores over $|M|$ modules. In turn, the generator $\Phi \in \mathbb{R}^{d \times |M|}$ can be considered a matrix of module parameters stacked column-wise, where each module has $d$ parameters. Thus, the generated parameters $\theta_t$ is a linear combination of the columns of the linear generator. This is also reminiscent of tensor factorisation models where parameters are factorised into shared tensors and task-specific tensors (Yang & Hospedales, 2017), which in hypernetworks correspond to the generator and the task embedding, respectively. However, hypernetworks learn both sets of parameters jointly rather than obtaining them from a factorisation of task-specific networks *a posteriori*.

### 4.3  Level of Routing

Another aspect of designing a routing function is its level of granularity. Routing can select modules globally for the entire network, make different allocation decisions per layer, or even hierarchically select sub-routers. This last method is also referred to as 'dispatched routing' by Rosenbaum et al. (2018). A naive version of global routing (Figure 5b) assumes that a single routing configuration is shared across layers. Allowing for different decisions per layer (Figure 5a) is more challenging as the space of potential architectures grows exponentially as $|M|^l$, where $l$ is the number of layers or sub-functions of the network. In fact, to compute the posterior over parameters, one would need to marginalise over every possible configuration of $A = [\alpha_1, \ldots, \alpha_l]$. Kirsch et al. (2018) resort to Expectation Maximisation to make it tractable. Instead, per-layer routing (Figure 5c) assumes conditional independence among decisions, thus facilitating scaling. Crucially, routing scores are sometimes employed not only to select a subset of modules but also to aggregate their outputs. This second purpose is addressed in more depth in § 5.

Most methods assume that routing decisions occur in a sequence, whose length is bounded or unbounded. This is the case where the output of every layer is fed into the next. However, routing may also involve defining both the selection of modules and their order of composition (i.e., the model architecture). For instance, in Neural Module Networks (NMNs; Andreas et al., 2016b; 2017), the routing function consists of a parser that takes in a query and produces a dependency tree. This is post-processed and transformed into a tree graph where nodes are modules and directed edges control the flow of the information, i.e. route the output(s) of a subset of modules as input to another module. In Modular Meta Learning, Alet et al. (2018) alternate between sampling compositional graphs using simulated annealing (Kirkpatrick et al., 1983) and performing a step of gradient descent on the network parameters for a set of meta-training tasks.

- The Routing Function is a critical component in modular neural networks, responsible for determining how information flows through modules.
- Routing can be categorized into Fixed Routing, Learned Routing, and Hypernetworks.
  - **Fixed Routing** uses predetermined rules to direct information flow.
  - **Learned Routing** employs neural networks to dynamically allocate modules based on input data or task information.
    * Challenges in Learned Routing include Training Instability, Module Collapse, and Overfitting, which require specialized strategies for mitigation.
    * **Hard Learned Routing** involves discrete module selection, often requiring reinforcement learning or stochastic re-parameterization for training.
    * **Soft Learned Routing** often use weighted combinations of modules, where predominantly top-$k$ routing strategies are employed for computational efficiency.
  - **Hypernetworks** offer a flexible approach by generating task-specific parameters with unnormalized routing scores.
- Routing decisions can occur at different levels of granularity, including global, per-layer, and hierarchical routing.

## 5 Aggregation Function

While in the previous section on *routing* we have covered the topic of how to *select* different modules during training, we will now focus on how we can *aggregate* these functions in order to combine the respective information. It is important to emphasise that, for the majority of current approaches, routing and aggregation are inseparable; that is, the *selection* and *aggregation* of modules are performed simultaneously.[17] On the other hand, the strategies for *aggregating* functions in this section are reminiscent of the taxonomy previously discussed for *computation* functions (see §3); while in the latter we looked into the composition of *shared* components with *modules*, in this section we provide insights into the composition of *multiple modules*. This is often required when modules are recombined for zero-shot transfer or task-level generalisation (for more details on these applications, see § 7).

In particular, for a subset of active modules $K \subseteq M_i$ the aggregation of modular components can (similarly) be realised on the *parameter* level $f'_i(\boldsymbol{x}) = f_{\boldsymbol{\phi}_1 \oplus \cdots \oplus \boldsymbol{\phi}_{|K|}}(\boldsymbol{x})$, *input* level $f'_i(\boldsymbol{x}) = f_{\boldsymbol{\theta}_i}([\boldsymbol{\phi}_1, \ldots, \boldsymbol{\phi}_{|K|}, \boldsymbol{x}])$, as well as *function* level $f'_i(\boldsymbol{x}) = f_{\boldsymbol{\phi}_1} \circ \ldots \circ f_{\boldsymbol{\phi}_{|K|}}(\boldsymbol{x})$. In addition, we cover *output* or *representation* level aggregation $f'_i(\boldsymbol{x}) = f_{\boldsymbol{\theta}_i}(\boldsymbol{x}) \oplus f_{\boldsymbol{\phi}_1}(\boldsymbol{x}) \oplus \cdots \oplus f_{\boldsymbol{\phi}_{|K|}}(\boldsymbol{x})$. Crucially, this differs from parameter aggregation if $f$ is non-linear. We discuss these different strategies in the following sections.

### 5.1 Parameter Aggregation

**Mode Connectivity** A natural strategy to aggregate information from multiple modules is interpolating their weights. However, given that neural *architectures* differ, and that hidden representations might not necessarily be equivalent (e.g. under invariance to invertible linear transformations) even if the model architectures are the same (Kornblith et al., 2019), naively aggregating module weights may have catastrophic consequences. However, recent work on *linear mode connectivity* (Frankle et al., 2020) suggests that under certain conditions, it is in fact possible to interpolate between multiple models, which has positive ramifications for modular aggregation methods. To understand these conditions, we first provide a brief introduction to the constraints under which parameter aggregation is permissible.

The phenomenon where the minima found by two networks are connected by a path of non-increasing error, has been the subject of research for many years (Freeman & Bruna, 2017; Draxler et al., 2018; Garipov et al., 2018; Nagarajan & Kolter, 2019). However, most works demonstrate that mode paths are in fact

---

[17]Combining modules has the potential to significantly improve inference speed.

not linear. While Nagarajan & Kolter (2019) find linear paths between networks, their experimental setup requires initialising models with the same set of weights. Frankle et al. (2020) and Neyshabur et al. (2020) demonstrate that this *linear* mode connectivity phenomenon is closely linked to the *Lottery Ticket Hypothesis* (Frankle & Carbin, 2019), which suggests that only a small subset of randomly initialised weights are the main drivers for the final performance of a model—the so-called *winning tickets* (see § 3.1). When interpolating between models trained on different tasks but initialised with the same set of weights, the models tend to stay in the same loss basin, indicated by the lack of a sudden increase in loss when interpolating the weights. Consequently, it appears that the flatness of the basin of the loss landscape translates to better generalisation capabilities of a model. Gueta et al. (2023) find that fine-tuned models reside in distinct regions in weight space, and models within the same region exhibit high performance. On the other hand, Ainsworth et al. (2022) argue that the success of such interpolation is strongly connected to the inherent bias of the optimiser being used, and not the neural network architecture itself.

**Weight Interpolation**   Building on the findings of interpolating the weights of models, Ansell et al. (2022) propose Lottery Ticket Sparse Fine-Tuning (LT-SFT), described in § 3.1. In particular, they identify language, and task-specific sub-networks $\phi_l$ and $\phi_t$. These can be aggregated by simply adding them to the base model parameters, i.e. $\theta' = \theta_0 + \phi_l + \phi_t$. Instead of identifying task adaptations on subsets of model parameters, Ilharco et al. (2022) propose to edit entire models with further arithmetic operations. For example, for toxic language generation and language modelling tasks, by performing the arithmetic negation operation $\theta' = \theta_0 + (\phi_{\text{general}} - \phi_{\text{toxic}})$, their new model $f_{\theta'}(x)$ generates less toxic text. This idea was influenced by the word analogy task (i.e., 'word arithmetics') (Mikolov et al., 2013).[18]

Rather than interpolating sparse adapters, Asai et al. (2022) aggregate parameters of soft prompts learned via prefix tuning (§ 3.2). In order to generalise to new tasks, (frozen) modules from past tasks and a learnable module created for the new task are interpolated according to the weights of an attention mechanism between the modules and the input.

**Model Merging**   Mode connectivity has enabled the fusion of entire models without extensive retraining, yielding performance improvements across a range of applications (Choshen et al., 2022; Gupta et al., 2020; Yadav et al., 2023; Jin et al., 2023). These developments have made frameworks, such as Git-Theta (Kandpal et al., 2023), which facilitate collaborative model development through version control, reasonable. Soft Merging of Experts with Adaptive Routing (SMEAR) (Muqeeth et al., 2023) introduces gradient-based training for sparsely activated models, offering specialization benefits.

## 5.2   Representation Aggregation

Closely related to parameter aggregation, representation aggregation consists of interpolating the outputs of individual modules. Crucially, both operations are equivalent if the functions are linear: $(\alpha_i \Phi_i + \alpha_j \Phi_j)x = \alpha_i \Phi_i x + \alpha_j \Phi_j x$. However, this does not hold true for non-linear functions, e.g. if the module is an adapter layer (Houlsby et al., 2019) or a feed-forward component of a Transformer layer (Fedus et al., 2021).

**Weighted Representation Averaging**   At the $i$-th sub-function of the model, where multiple modules $\phi \in M_i$ exist, the representations are passed through the (active) modules, outputting $|K_i|$ (latent) representations $h_1, \ldots, h_{|K_i|}$. One way of performing aggregation is to learn the weights $\alpha$ to interpolate over the hidden representations:

$$f_i'(x) = \sum_j^{|K_i|} \alpha_j h_j \tag{8}$$

with $\alpha_j$ being a module-specific scalar weighting.

This aggregation is equivalent to Equation (6) when interpreting each weight $\alpha_j \in [0, 1]$ as the output of a soft router, i.e. $\alpha_j = r(\phi_j)$. Consequently, all soft-learned routing approaches (e.g. MoE) that do not perform top-1 routing (see § 4.2.3) also determine how to aggregate the representations of different modules.

---

[18]$vec$('King') $- vec$('Man') $+ vec$('Woman') $\approx vec$('Queen'), with $vec(\cdot)$ denoting word embeddings of the respective words.

As an extension to the traditional MoE aggregation/routing function, Ma et al. (2018) propose to learn one aggregation function per task $t$ in a multi-task setup. Gururangan et al. (2022) pre-train modular components for different textual domains $d \in \mathcal{D}$. When utilising the pre-trained modules on unseen data, they weight the output representations $\mathbf{h}_d$ of the respective domain modules $\phi_d$ according to the posterior distribution over the input examples, i.e. $\boldsymbol{\alpha} = p(\mathcal{D} \mid \mathbf{x})$:

$$f'_i(\boldsymbol{x}) = \sum_{d \in \mathcal{D}} p(d \mid \mathbf{x})\, f_{\phi_d}(\boldsymbol{x}) \tag{9}$$

This posterior is inferred through the Bayes rule. This does not require any auxiliary model, and only relies on the original $d$-conditioned language model. In fixed routing, module representations are often averaged without weighting (Zhang et al., 2022a; Chronopoulou et al., 2022a). Similarly, in hard routing methods, the representations of all *active* modules are averaged, such as in Polytropon (Ponti et al., 2022), or summed, as in PathNet (Fernando et al., 2017).[19]

One disadvantage of simply learning gating parameters is that the weights do not depend on the hidden representations. Thus, they do not take into account their information content. This issue is tackled by *attention-based aggregation* functions.

**Attention-Based Representation Aggregation** Instead of inferring the weighting before a module has performed its transformation on the latent representation, the aggregation decision can take place *afterwards*. This allows for identifying whether or not the information added by the respective module is ancillary to the target task. In AdapterFusion, Pfeiffer et al. (2021a) propose an attention mechanism (Bahdanau et al., 2015) between the stacked hidden representations $\boldsymbol{H}_i$ produced by the modules and their input $\boldsymbol{x}$:

$$f_i(\boldsymbol{x}) = \mathrm{Attn}(\boldsymbol{x}\boldsymbol{Q}_i, \boldsymbol{H}_i\boldsymbol{K}_i, \boldsymbol{H}_i\boldsymbol{V}_i) \tag{10}$$

where $\boldsymbol{Q}, \boldsymbol{K}, \boldsymbol{V} \in \mathbb{R}^{d \times h}$ are the projections that produce the queries, keys, and values, and $\boldsymbol{x}$ is the input representation *to* each of the modules (i.e., the output representation of the previous layer). $\boldsymbol{H}_i \in \mathbb{R}^{|M| \times d}$ is a matrix consisting of row-wise stacking of the output representations $\boldsymbol{h}_1, \dots, \boldsymbol{h}_{|M_i|}$ of each module. In other words, the input of each module is interpreted as the query and the output of each module is interpreted as the value and key. The attention mechanism thus learns to attend over the module representations and weigh them according to their relevance for the current task.

Instead of aggregating module outputs into a single representation, Recurrent Independent Mechanisms (Goyal et al., 2021) concatenate the outputs of the top-$k$ active modules. However, in between the application of recurrent computation functions, they exploit an attention mechanism over hidden representations to enable sparse communication among modules.

One major disadvantage of both *weighted* and *attention-based* representation averaging, is that—when used in combination with soft routing—they require a full forward pass through all modules, even if they contribute only marginally to the final aggregation. Thus, they incur significant increases in time and space complexity. While this can be mitigated by pruning (i.e., dropping) some modules during inference (Rücklé et al., 2021), latency still remains an issue for scalability. Thus, top-$k$ hard routing offers a more efficient solution for both weighted averaging (Shazeer et al., 2017; Lepikhin et al., 2021) and attention-based aggregation (Goyal et al., 2021).

## 5.3 Input Aggregation

Input aggregation lends itself naturally to adapters such as prompts or prefix tuning (Brown et al., 2020; Lester et al., 2021; Li & Liang, 2021, see § 3.2). In prompting, we have a set of instructions or few-shot examples $\phi_1, \dots, \phi_{|K|}$. Given that the nature of prompting is to prepend the prompts to the input, aggregating the respective modules boils down to concatenating all prompts. That is, providing the model with *multiple*

---

[19]Note that the latter strategy leads to high variance in the norms of hidden representations if the router can select variable-size subsets of modules.

instructions, or with multiple examples (i.e. few-shot in-context learning) is a version of module input aggregation $f_1'(\boldsymbol{x}) = f_{\boldsymbol{\theta}_1}([\boldsymbol{\phi}_1, \ldots, \boldsymbol{\phi}_{|K|}, \boldsymbol{x}])$. This concept also extends to prefix-tuning, where we can simply concatenate all prefixes at every layer: $f_i'(\boldsymbol{x}) = f_{\boldsymbol{\theta}_i}([\boldsymbol{\phi}_i^1, \ldots, \boldsymbol{\phi}_i^{|K|}, \boldsymbol{x}])$.

In the context of prompting, Schick et al. (2021) leverage input aggregation by concatenating multiple textual descriptions of undesired behaviours of a language model to generate toxic text for model debiasing. In the context of prompt tuning, Vu et al. (2022b) learn separate task and language soft prompts that are recombined for zero-shot cross-lingual transfer in summarization. Nayak et al. (2022) compose soft prompts of attributes and objects in visual tasks to generalise to new classes. Soft prompts can also be aggregated with methods different from concatenation such as attention-based parameter interpolation (Asai et al., 2022).

Furthermore, input aggregation methods have found significant utility in retrieval augmented generation (Lewis et al., 2020a), a technique where retrieval models are employed to retrieve external knowledge for addressing knowledge-intensive NLP tasks.[20] In RAG methods, retrieved documents are appended to the input, essentially aggregating external information with the model's input. These methods facilitate knowledge injection and editing (Verga et al., 2021; Cheng et al., 2023), allowing models to access and incorporate information from external sources, which can be crucial for tasks demanding domain-specific knowledge or real-time data updates. This aligns with the broader theme of knowledge enhancement within modular neural architectures, extending their capabilities to handle complex and dynamic information needs.[21]

**Hypernetworks**  Similarly to soft prompts, hypernetworks may aggregate information from different embeddings by combining them in the input to the parameter generator. For instance, in (Ponti et al., 2021) task and language embeddings are concatenated in the input when training a multilingual multi-task architecture where the encoder is fully shared and the hypernetwork generates the classifier head. By recombining embeddings appropriately, this method allows for inferring the parameters of unseen task–language combinations. Combinations of embeddings have been used to generate adapters in multilingual (Üstün et al., 2020) and multi-task settings (Mahabadi et al., 2021b; Pilault et al., 2021).

### 5.4  Function Aggregation

Finally, aggregation can be achieved on the function level; $f_i'(\boldsymbol{x}) = f_{\boldsymbol{\phi}_1} \circ f_{\boldsymbol{\phi}_2}(\boldsymbol{x})$. Different aggregation methods infer either a sequence or a (tree) structure that determines the order of the aggregation.

**Sequential Aggregation**  By performing a forward pass through multiple modules, where the input to the next module is the output of the previous one, the respective hidden representations are sequentially transformed: $f_i'(\boldsymbol{x}) = f_{\boldsymbol{\phi}_1}(f_{\boldsymbol{\phi}_2}(\ldots(f_{\boldsymbol{\phi}_{|M|}}(\boldsymbol{x}))))$.

This form of information aggregation is often chosen in conjunction with *fixed routing*, as discussed in § 4.1, given that the routing order is determined by the role of each module (e.g. language and task adapters). Pfeiffer et al. (2020b; 2021b) propose a two-stage setup where language-specific components are disentangled from task-specific components, in order to perform zero-shot cross-lingual transfer. First, language (adapter) modules $f_{\boldsymbol{\phi}_{l_s}}$ and $f_{\boldsymbol{\phi}_{l_t}}$ are trained on monolingual unlabelled data for the source language $s$ and the target language $t$, respectively. Then, in the second stage, the language component $f_{\boldsymbol{\phi}_{l_s}}$ is inserted but frozen, and a new (adapter) module is added for a task $f_{\boldsymbol{\phi}_t}$ and trained on annotated data for the source language: $f_{\boldsymbol{\phi}_t}(f_{\boldsymbol{\phi}_{l_s}}(\boldsymbol{x}))$. Since this effectively disentangles language from task information, this also enables zero-shot inference on the target language $t$ without annotated data. In particular, $f_{\boldsymbol{\phi}_{l_s}}$ is substituted with $f_{\boldsymbol{\phi}_{l_t}}$, thereby hierarchically aggregating the information from the respective modular components: $f_{\boldsymbol{\phi}_t}(f_{\boldsymbol{\phi}_{l_t}}(\boldsymbol{x}))$. Similarly, Stickland et al. (2021) perform function composition of a language module $f_{\boldsymbol{\phi}_l}$ and a domain module $f_{\boldsymbol{\phi}_d}$ for multilingual multi-domain machine translation. For more examples, see § 7.1.

**Hierarchical Aggregation**  Alternatively, when global routers jointly determine the selection of modules and the model architecture, the order of function composition follows the structure of a tree. For instance,

---

[20]Notably, RAG is also discussed in § 3.2 due to its dual capability of both input composition and aggregation, for instance when multiple documents are used in the retrieval process.

[21]For further applications of RAG see § 7.1.4.

Neural Module Networks (Andreas et al., 2016b) leverage a semantic parse to infer a graphical structure for module aggregation. While all leaf nodes find objects by identifying regions of an image through attention, intermediate nodes either transform or combine these representations (depending on the arity of the node). The root then predicts the label by describing or measuring the attended objects.

> - Aggregation functions play a crucial role in combining information from multiple modules in modular neural architectures.
> - **Parameter aggregation** strategies interpolate weights of multiple modules and are influenced by concepts like linear mode connectivity.
> - **Weighted representation averaging** and **attention-based aggregation** are methods for combining the outputs of modules, with attention mechanisms allowing for dynamic weighting based on relevance.
> - **Input aggregation** methods, such as prompting and prefix tuning, involve concatenating instructions or examples to the input, enabling modular control over tasks and domains.
> - **Hypernetworks** can also perform input aggregation by combining different embeddings.
> - **Function aggregation** occurs on the function level and can be sequential or hierarchical, with different methods determining the order of aggregation.
> - Sequential aggregation is often used with fixed routing, while hierarchical aggregation is employed when global routers jointly determine module selection and model architecture.

## 6 Training Setting

Finally, we explore the training settings for modular architectures. We can identify three main strategies in the literature: **1)** all modules are jointly trained for *multi-task learning*; **2)** modules are introduced at different stages during *continual learning*; and **3)** in *transfer learning*, modules are added *post-hoc* after pre-training, often as a way to fine-tune the model in a parameter-efficient fashion. Importantly, these strategies are not necessarily mutually exclusive and can be realised in combination.

### 6.1 Joint Multitask Learning

In joint multi-task learning, there are two main settings. Firstly, task-specific parameterised components can be integrated into shared neural network architectures as a means to mitigate catastrophic forgetting or negative interference (McCloskey & Cohen, 1989; French, 1999) and as a way to scale the model size efficiently (Kudugunta et al., 2021). In these scenarios, modules are often optimised on individual tasks via fixed routing and specialise accordingly (Hampshire & Waibel, 1992; Rajendran et al., 2017, *inter alia*; see § 4.1 for more details). As an alternative, the architecture can be fully modular, sharing only the parameters for learned routing (Jacobs et al., 1991b;a; Rosenbaum et al., 2018; Kirsch et al., 2018; Chang et al., 2019, *inter alia*; see § 4.2.3 for more details).

Joint training can also be performed before *post-hoc* training: a shared base model can be pre-trained on multiple tasks as a warm-up before creating task-specific sparse subnetworks (Sun et al., 2020a) or as a way to provide a useful initialisation for modular parameters (Vu et al., 2022c). Dua et al. (2022) convert a dense language model pre-trained on text data into an MoE by decomposing the learned feed-forward layers. Pfeiffer et al. (2022b) add language-specific layers during multilingual pre-training of a language model. This prepares the model to be extended to more languages *post-hoc*; when new languages become available, a new (randomly initialised) learnable layer can be added to the inventory of modules, whereas the shared parameters remain untouched.

## 6.2 Continual Learning

In a similar vein to countering negative interference in multi-task learning, continual learning—that is, continuously integrating new data into the model—often aims at mitigating catastrophic forgetting (i.e., the knowledge learned at early stages of training should not get overwritten by updates to the model later on).

Similar to the multi-task learning approaches discussed in § 6.1, new layers can be continuously introduced within the network which are only updated on the new data, keeping the others untouched. In methods like Progressive Networks (Rusu et al., 2016), PathNet (Fernando et al., 2017), and PackNet (Mallya & Lazebnik, 2018) when the model is trained on a new task, the parameters of the previous tasks are frozen; however, for new tasks, new modules may be learned, which connect to the existing set of modules. Often, the decision of inserting new modules at a given stage is made dynamically based on outlier detection (Ostapenko et al., 2021). Progressive Networks (Rusu et al., 2016), on the other hand, scale the model capacity linearly with the number of tasks. Aljundi et al. (2017) train separate experts for every task and route new examples based on the distribution of the reconstruction errors of task-specific auto-encoders.

Instead of adding new parameters to the model, other works in the continual learning landscape identify subnetworks for different tasks. For instance, some works identify subnetworks of the model, which have not been used by previous tasks. Consequently, updating these parts of the model will have little effect on the previously learned knowledge (Javaloy & Valera, 2022). Similarly, 'supermasks' (§3.1; Wortsman et al., 2020), which learn a binary mask over a randomly initialised model, enable the extension to a potentially vast number of tasks during continual learning. Supermasks of previous tasks can be also linearly combined as a way to generalise to new tasks.

## 6.3 Parameter-efficient Transfer Learning

Recently, transfer learning has become the dominant strategy for state-of-the-art results on most tasks. Auxiliary self-supervised objectives are utilised to pre-train models on a large amount of data. Subsequently, the model's weights are fine-tuned on the target tasks (Howard & Ruder, 2018; Devlin et al., 2019). Updating a small set of parameters of these large models has been demonstrated to perform equally well as full model fine-tuning, leading to the emergence of parameter-efficient fine-tuning strategies.

Most methods discussed in § 3 that are applied to large pre-trained models can be considered as post-hoc adaptation. Modularity can be achieved through **parameter composition** (§ 3.1) using *sparse subnetworks* (Mehta, 2019; Chen et al., 2020; Donahue et al., 2014; Cai et al., 2020; Ben Zaken et al., 2022; Guo et al., 2021), or *low-rank adapters* (Li et al., 2018; Hu et al., 2022), **input composition** (§ 3.2) by augmenting the function's input (Brown et al., 2020; Li & Liang, 2021), and **function composition** (§ 3.3) through *adapter layers* (Rebuffi et al., 2017; Houlsby et al., 2019) and *rescaling* (Liu et al., 2022b). Additionally, hypernetworks can be used to generate the parameters of any of the above-mentioned types of modules (§ 3.4). Essentially, all of these methods are tightly connected as they share the same functional form (§ 3.5).

> - There are three main training strategies: (1) Joint Multitask Learning, (2) Continual Learning, and (3) Parameter-efficient Transfer Learning.
> - In **Joint Multitask Learning**, task-specific components are integrated into shared neural architectures, allowing modules to specialize via fixed or learned routing.
> - **Continual Learning** methods aim to integrate new data while mitigating catastrophic forgetting, with options to introduce new modules dynamically or identify subnetworks for different tasks.
> - **Parameter-efficient Transfer Learning** involves pre-training models on large datasets and fine-tuning on target tasks. Modular strategies can be applied post-hoc through various composition methods, including parameter composition, input composition, and function composition.
> - These training strategies are not mutually exclusive and can be combined to achieve specific goals in modular neural architectures.

# 7 Applications in Transfer Learning

Most applications of modular deep learning revolve around transfer learning. In particular, the two main purposes are: **1**) *parameter-efficient* fine-tuning (§ 7.1), which achieves superior efficiency, prevents negative interference, and enables zero-shot transfer; and **2**) zero/few-shot generalisation to new tasks (§ 7.2). In what follows, we provide a quick overview of transfer learning applications of modular deep learning. For the in-depth discussions and illustrations of the key concepts, we will first focus on applications in NLP, and then draw direct analogies with other deep learning areas such as speech processing, computer vision, and multi-modal (representation) learning. In § 8, we will explore additional purposes of modular deep learning, including hierarchical reinforcement learning, programme simulation, and causal inference.

## 7.1 Parameter-Efficient Fine-tuning

Regardless of the application area, one of the principal uses of modules has been to boost parameter efficiency and decrease model storage requirements of fine-tuning, eschewing so-called *full model fine-tuning* which requires storing a separate copy of the full model per task (Howard & Ruder, 2018; Devlin et al., 2019), see §6.3. In the simplest formulation, all task-specific updates are pushed to the parameters of the lightweight modules, while the parameters of the large base model are kept *frozen* throughout task fine-tuning. The modules then store *task-specific knowledge* that can be composed with the 'general-purpose' knowledge of the base model to adapt it to the task at hand. In NLP, this led to a number of research papers that introduced diverse modular architectures, as surveyed in § 3 and § 6. A typical evaluation protocol is fine-tuning a type of module on the popular GLUE and SuperGLUE benchmarks (Wang et al., 2019), comparing against full model fine-tuning or alternative modular architectures. The results usually corroborate either of two main goals: (i) improving performance with the same parameter budget versus (ii) maintaining performance with a smaller parameter budget (Mahabadi et al., 2021a; Zhou et al., 2023). In addition, modular adaptation has further benefits: first, it prevents negative interference between tasks (Bapna & Firat, 2019). Second, it allows for combining adapters to enable zero-shot transfer (Pfeiffer et al., 2020b). In light of the enormous size of state-of-the-art large language models (LLMs), parameter-efficient fine-tuning has emerged as the main way to update the pretrained models (Hu et al., 2022).

### 7.1.1 Machine Translation

In the seminal work of Bapna & Firat (2019), *bilingual* (i.e., language-pair) adapters (see §3.3) were used to adapt a massively multilingual NMT model (spanning 103 languages) to a particular source–target translation direction. One benefit of such bilingual adapters is their ability to 'skew' the multilingual model to the language pair at hand without losing the benefits of massively multilingual training for low-resource languages. Another positive effect of bilingual adapters concerns recovering the MT performance also for high-resource languages. High-resource languages might typically suffer from performance deterioration due to the particular interference phenomenon known as the 'curse of multilinguality' (Conneau et al., 2020; Wang et al., 2020): when (too) many languages compete for the fixed parameter budget of the model, the model's expressiveness and representation power deteriorates for all languages. The use of modules extends the parameter budget to recover the detrimental effects of multilingual inference through dedicated (i.e., modular) bilingual adaptation. Their work also demonstrates the superior performance of a multilingual model specialised towards a particular language pair over merely training a bilingual NMT model for the same pair from scratch.

However, fine-tuning bilingual adapters (or more generally, modules) for each translation direction assumes parallel data for all language pairs and requires $n(n-1)$ modules to cater for all possible language pairs (one dedicated module in the encoder and another module in the decoder). Therefore, follow-up work (Philip et al., 2020; Üstün et al., 2021) aimed to learn *monolingual* (i.e., language-specific) adapters. Again assuming standard encoder-decoder architectures for MT such as mBART (Liu et al., 2020), this design requires only $2n$ modules in total. Besides improving parameter efficiency, this also bypasses the critical dependency on parallel data for *all* language pairs and allows for learning from monolingual data. Crucially, this design also enables translation to or from languages without parallel data, in a fully unsupervised way, and even to/from languages unseen by the base pre-trained encoder-decoder model. Put simply, when translating from language $l_s$ to $l_t$, only the encoder adapters for $l_s$ plus the decoder adapters for $l_t$ are activated: the model is

able to translate from $l_s$ to $l_t$ without seeing a single parallel $l_s$ to $l_t$ sentence. This application in the field of NMT exemplifies the power of modular design: available components, which were previously learned locally and asynchronously, can be recombined in novel ways to generalise systematically to unseen applications (i.e., in this particular case, to unseen translation directions). This is one of the main goals of modular deep learning (§ 1).

The separation into dedicated language-specific modules mitigates interference and catastrophic forgetting; however, it also hinders any positive transfer between modules of similar languages. The positive transfer can be achieved through the use of hypernetworks (see §3.4): Baziotis et al. (2022) learn to generate monolingual language-specific adapters for NMT. In fact, sharing the parameter generator takes advantage of language similarities (Platanios et al., 2018). As discussed in more detail later in §7.1.2, similar ideas of combining the modular design with hypernetworks have also been applied earlier and beyond NMT, e.g., for task fine-tuning with adapters in monolingual multi-task setups (Mahabadi et al., 2021b) and for cross-lingual transfer in single-task (Ansell et al., 2021) as well as in multi-task setups (Ponti et al., 2021; Üstün et al., 2022).

The curse of multilinguality and catastrophic interference in multilingual MT models have also been tackled through sparse sub-networks (see § 3.1). Lin et al. (2021) extract sparse sub-networks for specific language pairs from a trained multilingual MT model via pruning. Subnetworks are then trained separately in order to specialise towards the particular translation direction. In fact, there exist dedicated small sub-networks (which can be obtained via standard masking) that store language pair-specific knowledge within the large network, where such knowledge should not interfere with other language pair-specific sub-networks (Dua et al., 2022). The same high-level idea has also been applied to *domain adaptation of bilingual MT systems*: e.g., Liang et al. (2021) show that it is possible to learn domain-specific sub-networks when fine-tuning the MT system on new domains, where a single large network (i.e., the full neural MT system) comprises multiple disjoint domain-specific sub-networks specialised to particular domains.

Another approach that leverages modularity for an increased language-specific capacity in MT is mixture-of-experts. Each expert is typically dedicated to a particular language or translation direction (Kudugunta et al., 2021; Costa-jussà et al., 2022). To maintain feasible decoding time, the procedure works as follows: (i) during training, mix the inputs from different translation directions in the same batch, in order to learn the routing network and encourage positive transfer among related tasks; (ii) at inference time, different translation directions are decoded separately, and only the corresponding subset for elevant experts is loaded.

### 7.1.2 Cross-Lingual Transfer

NMT focuses on translation as a single task and modularity was exploited mainly to carve language-specific and/or domain-specific modules that can support multilingual and multi-domain systems, respectively. In more general cross-lingual transfer setups, the aim is to transfer large models (Devlin et al., 2019; Conneau et al., 2020) fine-tuned *for any task* (e.g., sequence labelling tasks such as NER, text classification tasks such as NLI, sentiment analysis or intent detection for dialogue systems) on one or more source languages (where such task annotations exist) to one or more target languages (Hu et al., 2020; Ruder et al., 2021). Ideally, the transfer should be achieved without fine-tuning the full model (Hu et al., 2020), which results in catastrophic forgetting and negative interference, or requires the creation of separate model copies for each task.

The idea of training *language modules* thus largely follows what already outlined for MT in §7.1.1, with the addition of another set of dedicated modules that aim to capture task-related knowledge: *task modules*. Such language modules and task modules can then be combined to **1)** favour zero-shot cross-lingual transfer for particular source-target directions (Pfeiffer et al., 2020b; Ansell et al., 2021; 2022; Parović et al., 2022); **2)** provide extra capacity to low-resource languages under-represented (or even not covered) in the large multilingual models such as mBERT or XLM-R (Pfeiffer et al., 2021b; 2022b; Ponti et al., 2020; Faisal & Anastasopoulos, 2022), independently from task knowledge; and **3)** enable handling unseen language–task or even language–domain–task configurations (Ponti et al., 2021; Stickland et al., 2021).

As an example of zero-shot cross-lingual transfer, the original MAD-X framework (Pfeiffer et al., 2020b, Figure 1a) relies on bottleneck adapters to implement language and task modules: In particular: **1)** Language modules are inserted into each layer of the original neural model and are fine-tuned on (unsupervised) data of the particular language (e.g., via Masked Language Modelling) while the weights of the original model are kept

fixed. **2)** After obtaining language modules, task modules are *stacked* on top of the source language module(s) and are fine-tuned relying on the task objective and task-annotated data in the source language(s), while both the original model *and* language modules are kept fixed. **3)** At inference, source language module(s) are replaced with the desired target language module while retaining the task module: this enables zero-shot task inference in the target language.

Recent work has introduced a spectrum of variations and enhancements to this core idea. For instance, inspired by the bilingual 'translation direction' adapters for NMT systems (§7.1.1), Parović et al. (2022) learn bilingual adapters instead of single language adapters to boost transfer for a particular language pair. Faisal & Anastasopoulos (2022) and Chronopoulou et al. (2022b) learn language family adapters to reduce data sparsity for low-resource languages and capitalise on language similarity and cross-language sharing. Stickland et al. (2021) decouple language and domain knowledge into dedicated modules (see also §7.1.3 later). Further, Ansell et al. (2022) implement dedicated modules as sparse sub-networks, the so-called language and task masks, which can be composed with the base model via parameter composition. Following the analogy between language-specific and bilingual adapters, instead of learning separate language and task sub-networks, Foroutan et al. (2022) learn dedicated task–language sub-networks, demonstrating the variance in the extracted sub-networks across different task–language combinations. The use of such language sub-networks as language modules, even without dedicated task modules, improves cross-lingual transfer for dependency parsing when used within a meta-learning setup (Choenni et al., 2022). Litschko et al. (2022) compare sparse sub-networks and bottleneck adapters for transferring ranking functions for information retrieval tasks across languages and find them both superior to full model fine-tuning.

Finally, a body of work again focuses on 'contextually generating' the modules via hypernetworks, aiming to increase efficiency and benefit from connections between different languages *and* tasks. A representative example is the Hyper-X framework (Üstün et al., 2022) provided in Figure 6, where the module parameter generation is conditioned on the (disentangled) task and language, and additionally on the index of the Transformer layer where the generated module is inserted. Each task and language are parameterised via separate embeddings, which enables adaptation to any task–language combination, where these embeddings are low-dimensional

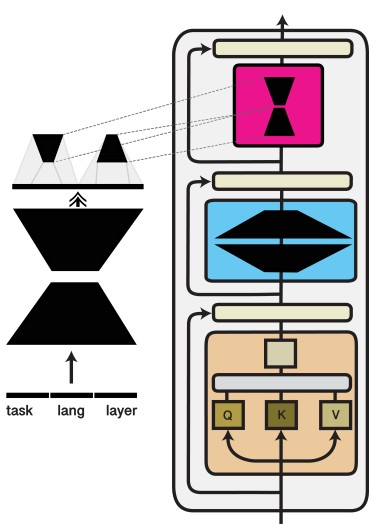

Figure 6: Hyper-X (Üstün et al., 2022): an example application of contextual module generation where a hypernetwork takes the concatenation of task, language and layer embeddings as input and generates a flat parameter vector. This is further reshaped into an adapter module within each Transformer layer. Learning independent layer embeddings and sharing a single hypernetwork across all layers (Ansell et al., 2021) (i) enables information sharing across layers, and (ii) reduces trainable parameters of the hyper-network by a factor corresponding to the number of layers.

vectors which are learned together with the parameters of the hypernetwork (see Figure 6 again). The framework thus leverages supervision and positive transfer from both multiple tasks and languages. Hyper-X can be seen as a more general variant of a series of precursors backed by the idea of contextual generation: Ponti et al. (2021) condition the hypernetwork on both task and language embeddings but generates only the model's classifier head. Other methods generate modules but condition the hypernetwork only on tasks in a monolingual setup (Mahabadi et al., 2021b) or only on languages in a cross-lingual transfer setup (Üstün et al., 2020; Ansell et al., 2021).

### 7.1.3 Domain Adaptation

As already hinted at in §7.1.1 and §7.1.2, dealing with different domains adds another tier to the modular design: domain-specific knowledge might be captured within dedicated *domain modules*.[22] This can again be accomplished through similar modular architectures as with language and task adapters. For instance, it is possible to inject domain-specific knowledge into (bottleneck) adapters (Zhang et al., 2021; Chronopoulou et al., 2022a) or to extract sparse domain-specific or task-specific sub-networks (Thompson et al., 2018; Ke et al., 2021b) for multi-domain and multi-task learning. Mixture-of-experts also enable multi-domain joint learning as well as domain adaptation (Guo et al., 2018; Zhong et al., 2022). Similar strategies have also been used in multi-domain and cross-domain speech processing and computer vision applications (see §7.1.5 and §7.1.6 later).

In domain adaptation, it is common to combine both shared parameters and domain modules that are learned jointly (Bousmalis et al., 2016). Beyond this standard setting, many approaches employ additional regularisation terms. The most common are **1)** a domain-adversarial loss on the shared parameters in order to encourage them to be domain-invariant (Ganin et al., 2016; Chen & Cardie, 2018); **2)** an orthogonality constraint on the domain modules to ensure that they capture different information (Baktashmotlagh et al., 2013; Kim et al., 2017); and **3)** similarity constraints that bring representations of similar modules close together (Bousmalis et al., 2016).

### 7.1.4 Knowledge Injection

Naturally, dedicated modules can also be assigned to inject and store external knowledge (e.g., from manually curated external knowledge bases), which can then interact with language, domain, or task knowledge. This idea has been explored with diverse external knowledge sources. For instance, Lauscher et al. (2020) aimed at complementing the distributional knowledge of large language models with conceptual and commonsense knowledge from ConceptNet (Speer et al., 2017). The external knowledge was captured within dedicated bottleneck adapters: they were fine-tuned via language modelling on synthetically created sentences from random walks over the ConceptNet graph structures. Majewska et al. (2021) stored verb-related knowledge from VerbNet (Schuler, 2005), a human-created verb classification repository, into bottleneck adapters, and demonstrated its usefulness in a range of tasks that require understanding of verb semantics. Along similar lines, Wang et al. (2021a) offered a generalisation of these approaches where different knowledge sources (e.g., Wikipedia, WikiData) are mapped to different dedicated adapters, which can be aggregated according to the task at hand. The same idea has been explored by Lu et al. (2021) in the biomedical domain, where the main knowledge sources were the UMLS Metathesaurus graph (Bodenreider, 2004) and biomedical Wikipedia articles. Lu et al. (2021) also introduce another component, the so-called knowledge controller, which can be seen as a standard attention-based function aggregator from §5.4. As an example of another relevant application, Lauscher et al. (2021) learned bottleneck adapters without manually curated external data, with the focus on model debiasing: the debiasing adapters were fine-tuned via standard language modelling on a counterfactually augmented corpus.

Finally, the idea of modular knowledge injection is also directly linked to the retrieval-augmented language models in text-only settings (Lewis et al., 2020b) as well as in multi-modal settings (Yasunaga et al., 2023) where the standalone retrieval module, detached from the 'main' language model, is responsible to fetch knowledge from some external memory or a knowledge base, and that knowledge is then used to condition the language model. In this design, the retrieval step and capability is made explicit and decoupled from the language model generation capability: as such, one can work directly on a retrieval module without the need to change the other components of the entire model (Yu et al., 2023). The ability of standard language models to use external tools is also sparked by the modular design: different external tools specialised for performing particular functions (e.g., conducting Web search, performing mathematical operations) are stored as separate modules accessed from the main model via external API calls. For a comprehensive overview of such *augmented language models*, we refer the reader to the recent survey Mialon et al. (2023).

---

[22]For instance, disentangling domain and language information yields benefits for NMT and cross-lingual transfer applications (Vilar, 2018; Cooper Stickland et al., 2021; Pham et al., 2021; Saunders, 2022).

### 7.1.5 Speech Processing

The use of modular deep learning for speech processing applications closely matches the ideas already exposed for NLP tasks. The landscape of the possible modular designs is exactly the same, where the only crucial differences are (i) the choice of the underlying large model, and (ii) the corresponding objective functions used to inject the specialised knowledge into the modules. For instance, the typical choice of the base model for automatic speech recognition (ASR) applications is one from the wav2vec family (Baevski et al., 2020; Babu et al., 2022), while the ASR-oriented objective function is the standard Connectionist Temporal Classification (CTC) loss (Graves et al., 2006). The high-level modular structure remains the same, as illustrated in Figure 7 with an example from Thomas et al. (2022), which utilises standard bottleneck adapters.

While in theory a large variety of possible modular configurations from § 3-§ 6 can be applied to diverse speech processing tasks, the majority of current work in the area has indeed focused on the use of bottleneck (sequentially placed) adapters for ASR in monolingual and multilingual contexts. Before that, the concept of modularity can be traced to the work of Swietojanski et al. (2016), where the model re-weights hidden units using small amounts of unsupervised data to better adapt to a particular speaker or an environment. More recently, bottleneck adapters have been used to perform ASR adaptation to atypical and accented speech (Tomanek et al., 2021), unseen speakers with limited adaptation data (Wang & Van hamme, 2022; Eeckt & Van hamme, 2022; Chen et al., 2023), new domains and manners of speaking (e.g., children's speech) (Fan & Alwan, 2022; Zhu et al., 2022), or to perform further model customisation to specific speakers (Biadsy et al., 2022; Sathyendra et al., 2022) and for multilingual learning (Kannan et al., 2019; Hou et al., 2022). A notable exception, not resorting to adapter layers, is the method of (Winata et al., 2020) which aims to learn low-rank modules (§ 3.1), akin to the idea of LoRA (Hu et al., 2022), for end-to-end ASR.

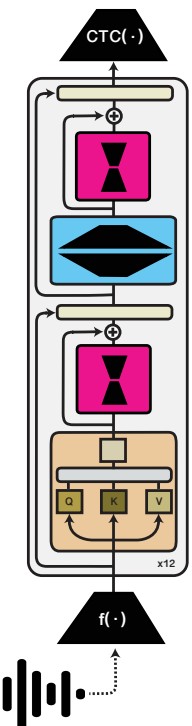

Figure 7: The structure of the wav2vec 2.0 model with task-specific bottleneck adapters for parameter-efficient ASR fine-tuning from Thomas et al. (2022); $f(\cdot)$ denotes a convolutional encoder followed by 12 standard Transformer encoder blocks. For downstream ASR a linear classifier, $CTC(\cdot)$, is applied to the final encoder output.

Multi-task (where the term 'task' in this context can e.g. refer to different languages, domains, speakers, or accents) ASR setups have also witnessed the usage of mixture-of-experts, closely following the basic ideas already discussed for NMT (§7.1.1) where different languages are assigned their dedicated modules through fixed routing. For instance, in speech processing, MoEs have been applied to multilingual ASR and cross-lingual ASR transfer (Bai et al., 2022; Gaur et al., 2021; Kumatani et al., 2021), while You et al. (2022) propose MoE for ASR with learned routing.

Beyond ASR, bottleneck adapters have also been used for speech translation (Le et al., 2021). Most recently, modular adapter-based approaches have been applied to text-to-speech methods (TTS) (Hsieh et al., 2022; Morioka et al., 2022), aiming to extend standard large multi-speaker TTS models such as FastPitch (Lancucki, 2021) to new speakers without compromising the TTS quality for the seen speakers. From a high-level perspective, one can see a direct analogy of this goal to the objectives in the MT literature of extending multilingual MT systems to unseen languages without compromising seen languages (see §7.1.1 again).

### 7.1.6 Computer Vision and Cross-Modal Learning

In computer vision, similar to NLP and speech processing (§7.1.5), dedicated modules are again used to enable parameter-efficient fine-tuning across multiple tasks and domains (Rusu et al., 2016; Rebuffi et al., 2018; Berriel et al., 2019; He et al., 2022b, *among others*). The core difference, again, is the choice of the actual neural architecture for the underlying model as well as for the modules: e.g., residual adapters (Rebuffi

et al., 2017) consisted of simple $1 \times 1$ convolutions combined with the base ResNet neural model (He et al., 2016) while other work learned task-specific convolutional filters (Newell et al., 2019; Bragman et al., 2019). More recent work aims to exploit modular architectures from NLP (e.g., sequential or parallel adapters, LoRA, prefix tuning) with pretrained Vision Transformer (ViT) architectures (Dosovitskiy et al., 2021): e.g., He et al. (2022b) run a comparative empirical analysis of various modular architectures for vision tasks, while Chen et al. (2021) rely on sparse sub-networks.

Modular design lends itself naturally to cross-modal and multi-modal applications, where different modalities may be captured by modality-specific parameters and routing can also be modality-conditioned. For instance, in multilingual vision-and-language (V&L) settings, it is possible to conduct inference in languages that lack labelled task examples. In fact, language knowledge is again disentangled from the task and modality knowledge, and the knowledge for different input modality streams can be captured in dedicated modules. This idea has been heavily explored in recent work in multi-modal multi-task scenarios, both in monolingual (Sung et al., 2022) and multilingual contexts (Bugliarello et al., 2022; Pfeiffer et al., 2022a), for tasks such as image captioning (Zhou et al., 2022a; Gao et al., 2021a), text-to-image generation (Maharana et al., 2022), visual question answering (Liu et al., 2022a; Sung et al., 2022), visual reasoning (Liu et al., 2021a), etc. For instance, Flamingo (Alayrac et al., 2022) uses frozen pretrained vision and language models, and only trains adapter layers to handle sequences of arbitrarily interleaved visual and textual data. It is trained with a sequence modelling objective on Web-scale data (Li et al., 2021) and displays impressive zero-shot and few-shot capabilities. Pfeiffer et al. (2022a) use adapter modules to equip multilingual text-only models with the ability to also process the visual modality, as well as to equip monolingual multi-modal models to deal with input from multiple languages. Papalampidi & Lapata (2022) rely on hierarchical adapters (akin to hierarchical representation aggregation discussed in § 5) for the task of summarising long videos into textual descriptions. Pan et al. (2022) demonstrate that modular design also helps in image-to-video transfer tasks: they use adapter modules to equip a large image-based model without temporal knowledge with the ability to reason about dynamic video content.

We note that in this survey, we aim to list some exemplary applications and draw parallels between different yet similar application areas such as NLP, speech processing, and computer vision. While we acknowledge that there exists a wealth of other work in these areas, we have no pretence of exhaustiveness.

### 7.1.7 Comparison and Design Principles

While a full-fledged comprehensive empirical study of the plethora of modular architectures across various application tasks and areas is still lacking, there exist initiatives such as the publicly available AdapterHub platform (Pfeiffer et al., 2020a): it provides (re)implementations of representative modular NLP architectures, within a unified framework tied to HuggingFace Transformers (Wolf et al., 2020). Among others, AdapterHub includes representatives of each computation method in § 3: LoRA (Hu et al., 2022) (i.e., low-rank parameter composition), prefix tuning of Li & Liang (2021) (input composition) and a number of bottleneck adapter configurations (function composition). The existence of AdapterHub delineates another crucial advantage of modularity: *reusability* of existing, already fine-tuned modules which can be (re)combined with the large neural models. In short, any practitioner can share or reuse a module specialised for a particular purpose (e.g., capturing specific task or language knowledge) with the community, facilitating community-wide sharing and thus avoiding time- and energy-costly repetitions of the same fine-tuning procedure.[23] As discussed in § 4, one can observe initiatives such as AdapterHub as continuously updating community-distributed multi-task models.

The discussion in this section also points to a more general principle: different end-goals even within the same end-application (e.g., NMT, cross-lingual transfer, domain adaptation) require rethinking the actual modular design, and the desired level and nature of modularity. For instance, if the goal in NMT (or cross-lingual transfer) is to boost performance for a particular translation or transfer direction, it might be useful to trade off some modularity for a better final performance by replacing language-specific monolingual modules with bilingual modules (Bapna & Firat, 2019; Parović et al., 2022). On the other hand, if the goal is to enable zero-shot or few-shot translation or transfer, the design with monolingual modules might be a better

---

[23]The (concept of) reusability enabled by the modular design also positively impacts energy consumption (Strubell et al., 2019), making an important leap towards Green(er) AI (Schwartz et al., 2020).

choice. In another example, if the focus is on MT or transfer for a particular low-resource language, the model designer should enable positive transfer to that language by 'opening' the flow of information from a module storing knowledge on high-resource languages similar to the target language if such languages exist (e.g., from Spanish to Galician) (Üstün et al., 2021), or by learning modules for families or groups of similar languages (Chronopoulou et al., 2022b; Faisal & Anastasopoulos, 2022). Analogously, related domains can also be grouped and hierarchically organised to enable positive transfer for domain adaptation (Chronopoulou et al., 2022a).

Other practical desiderata may also influence the selection of the actual modular design. If the final task performance is paramount, larger modules might be preferred, e.g., in order to offer enough extra capacity to store the wealth of language-specific information (Ansell et al., 2022). However, if model compactness is paramount, the criterion for choosing a specific design is instead the trade-off between efficiency (in terms of parameters and/or train and test time) and task performance; the optimisation of this trade-off has been the focus of recent research (Rücklé et al., 2021; Mahabadi et al., 2021a;b; Sun et al., 2022). In another example, if time efficiency during inference is a crucial requirement (e.g., real-time ASR in dialogue systems, low latency for information search systems) parameter composition methods such as sparse subnetworks or low-rank composition methods may be preferred over function composition methods as the latter increase the number of computations required during the forward pass, (see Table 3). In yet another example, if storage requirements are a critical constraint, one cannot resort to huge mixture-of-expert models where billions of parameters must be stored (Lepikhin et al., 2021).

## 7.2 Task Generalisation

The diverse applications of modular deep learning covered so far almost exclusively focus on learning modules associated with (arguably) well-formed and interpretable 'units of knowledge' such as languages, tasks, domains, dialects, accents, and speakers. However, modularity might also be achieved when such units are *unknown*. This relies on jointly learning arbitrarily sized inventories of so-called latent *skills* and a learned routing function (§ 4.2). Since such skills are learned end-to-end on a mixture of data from multiple tasks, they are often not straightforwardly interpretable. On the other hand, since arbitrary subsets of skills can be combined and each skill can be updated locally, these modular neural architectures are ideal for systematic generalisation to new tasks (Zhang et al., 2022a; Ponti et al., 2022).

In fact, another fundamental application in transfer learning is achieving zero-shot or few-shot generalisation to new tasks, where test examples are not i.i.d. with respect to training examples. The general experimental setup involves disjoint sets of training and test tasks. A model is pre-trained through multi-task learning on training tasks and then adapted to each new test task based on zero or few data points. Common examples of evaluation benchmarks for this setting include CrossFit (Ye et al., 2021), the T0 task suite (Sanh et al., 2022), or Natural Instructions (Mishra et al., 2022). While a common strategy to tackle this problem is instruction tuning (Sanh et al., 2022; Wei et al., 2022a), where models are fine-tuned prepending the instructions for each task, modular deep learning has emerged as a strong contender (Alet et al., 2018; Kudugunta et al., 2021; Ponti et al., 2022).

## 8   Other Purposes of Modularity

In addition to scaling large models (for instance, through MoEs, as discussed in § 4.2.3) and facilitating transfer learning, which we covered in § 7, modularity serves multiple additional purposes. In particular, we devote this section to a cursory view of modularity for i) hierarchical reinforcement learning (§ 8.1); ii) neural programme simulation (§ 8.2); iii) neural causal inference (§ 8.3). While most of these applications predate the advent of neural networks, (modular) deep learning expands the scope and potential of these lines of research for a series of reasons. First, it holds promise to induce the relevant latent structures (such as options, programmes, or causal graphs, respectively) in an end-to-end fashion. Second, it marries these traditional problems with the ability to jointly model low-level perception, such as vision and language.

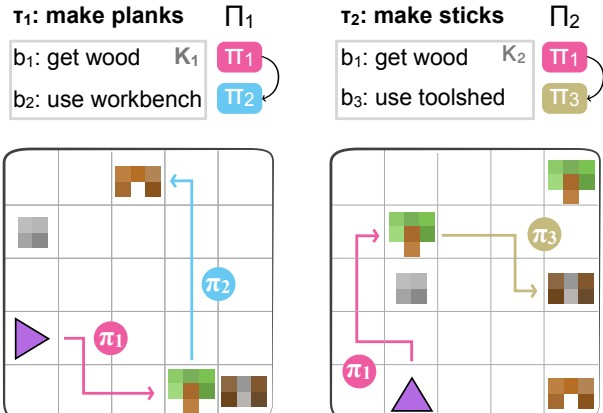

Figure 8: An example of **Hierarchical Reinforcement Learning** (§ 8.1), Policy Sketches (Andreas et al., 2017). Two high-level policies $\Pi$ corresponding to task instructions $\tau$ are illustrated. Each iteratively selects low-level policies $\pi$ (options) corresponding to sub-tasks $b$ from a shared inventory. These determine the choice of action given observations. In this case, options are implemented as predicate–argument pairs.

## 8.1 Hierarchical Reinforcement Learning

The goal of reinforcement learning is to learn a policy, which predicts the next action of an agent based on past observation(s) from the environment, that maximises the return, i.e. the sum of future discounted rewards. However, many tasks span extremely dilated temporal horizons or provide only highly sparse or delayed rewards. In these cases, it becomes helpful to model intermediate abstractions between high-level goal specifications and low-level actions and observations (Sutton et al., 1999; Precup, 2000). This facilitates the planning abilities of the agent as well as their sample efficiency. In fact, the above-mentioned intermediate abstractions, known as *options* or *skills*, consist of sub-policies that are transferable across tasks.

More formally, each reinforcement learning task is a Markov Decision Process (MDP) consisting of states $\mathcal{S}$ and actions $\mathcal{A}$, a transition function $p : \mathcal{S} \times \mathcal{A} \times \mathcal{S} \to [0, 1]$ and a reward function $r : \mathcal{S} \times \mathcal{A} \to \mathbb{R}$. We aim to learn a policy $\pi : \mathcal{S} \times \mathcal{A} \to [0, 1]$. We also define a value function as the expected (discounted) return from a given state $s$ as $V_\pi(s) = \mathbb{E}_\pi[\sum_{t=0}^{\infty} \gamma^t r_{t+1} \mid s_0 = s]$, as well as a Q function from a state $s$ and an action $a$ as $Q_\pi(s, a) = \mathbb{E}_\pi[\sum_{t=0}^{\infty} \gamma^t r_{t+1} \mid s_0 = s, a_0 = a]$. Following Sutton et al. (1999) and Precup (2000), each option $\omega \in \Omega$ is defined as a tuple $(\mathcal{I}_\omega, \pi_\omega, \beta_\omega)$, where $\mathcal{I}_\omega \subseteq \mathcal{S}$ is the initiation set, $\pi_\omega : \mathcal{S} \times \Omega \to [0, 1]$ the option-specific policy, and $\beta_\omega : \mathcal{S} \to [0, 1]$ is the termination function. For simplicity, many works assume that $\forall s \in \mathcal{S}, \forall \omega \in \Omega, s \in \mathcal{I}_\omega$: in other words, all options are available at every state. Augmenting a task with options transforms it into a Semi-MDP, with corresponding functions $\mathcal{V}_\Omega(\omega)$ and $\mathcal{Q}_\Omega(s, \omega)$.

Learning options involves a series of challenges (Jiang et al., 2019). Firstly, it is not trivial to specialise sub-policies towards distinct behaviours. This shortcoming is common to many modular architectures with learned routing (Mittal et al., 2022, see § 4.2). Not only this, the problem of hard learned routing has often been cast in a reinforcement learning framework (§ 4.2.2). Secondly, one must define the space where the actions of the high-level policy, which are latent variables, lie. In practice, one could treat them as a discrete, unordered set. In this case, a module from an inventory is chosen for a certain amount of time steps. However, alternative methods operate in structured spaces such as *language*, which is more transferable and scalable due to its combinatorial nature. Thirdly, training multiple options dilates the training time and requires collecting an appropriate amount of experiences for each of them. Fourthly, if trained jointly, options change simultaneously with the master policy, which is a source of non-stationarity. As a consequence, previous experiences for the master policy become invalid if the options have been updated in the meantime. Again, this is reminiscent of the challenges of learned routing exposed in § 4.2.

The simplest solution to circumvent end-to-end joint learning of the master policy and options is to provide *separate supervision* to both (Sutton et al., 1999; Dayan & Hinton, 1992). However, this may require extensive annotation, which is often not available. Thus, an alternative method is *defining sub-goals*, i.e. states an agent should reach as a stepping stone towards the high-level goal Dietterich (2000). Nevertheless, this similarly fails to scale due to the exponentially growing number of combinations of sub-goals some tasks may entail. Moreover, this does not eschew the need to train individual sub-policies for each sub-goal. A partial remedy is offered by *hindsight learning*, where an off-policy correction is introduced (Nachum et al., 2018). Specifically, the original target sub-goal of the current option is substituted with the one maximising the probability of the past sequence of low-level actions. Similarly, the master policy can be trained in hindsight through the currently predicted sequence of high-level sub-goals. Overall, relabelling past experiences significantly improves the model's sample efficiency.

A more radical solution to the challenge of scalability is jointly training both the master policy and options in an *end-to-end* fashion. To this end, Bacon et al. (2017) put forth a new architecture, the option–critic, that discovers options from data, without supervision for the intermediate abstractions. This architecture is trained based on policy gradient theorems Bacon et al. (2017) derive for options. Moreover, they augment the set of actions $\mathcal{A}$ available to each policy $\pi_\omega$ with a special end-of-policy action EOP instead of explicitly modelling $\beta_\omega$. Intuitively, formulating the execution as *call-and-return*, a master policy $\pi_\Omega$ determines the active option $\omega$, whose policy $\pi_\omega$ is followed until the EOP action is chosen. At this point, control returns to the master policy to choose the next option, and so on until termination. A downside of this method is that it is unstable and often diverges to degenerate solutions (Jiang et al., 2019). Thus, several inductive biases have been proposed to correct it. A popular method is leveraging *intrinsic rewards*: an auxiliary loss diversifies options by maximising the mutual information between each option and the next state conditioned on the current state (Florensa et al., 2017; Kulkarni et al., 2016).

An orthogonal question revolves around the ideal space for the option variables. In fact, compared to a discrete, unordered inventory of (possibly hard-coded) options, language affords more flexibility (Andreas et al., 2017; Jiang et al., 2019) as it solves many of the above-mentioned challenges. In fact, all sub-policies can be implemented through a single model conditioned on the linguistic label of the current option. This not only allows options to borrow statistical strength from each other but also makes options reusable in new tasks. Moreover, the nature of language (through its infinite use of finite means) is suitable to capture the extremely complex combination of sub-goals of many reinforcement learning tasks. Note that linguistic options can be interpreted as a generalisation of sub-goals, as every instruction implicitly corresponds to a subset of states (Jiang et al., 2019).

In practice, to learn linguistic options, Andreas et al. (2017) assumes that 'sketches' of options are provided for supervision (see Figure 8). To induce them, subsequent methods rely instead on synthetic experiences through relabelling (Jiang et al., 2019), or restricted vocabularies and syntax such as predicate–argument pairs (Das et al., 2018). Recently, the master policy has been frequently implemented as a large language model. Since these are pre-trained on text, they already contain world knowledge that can serve as a powerful inductive bias for grounded learning. For instance, Huang et al. (2022) use frozen language models to generate options through prompting in a zero-shot fashion.

## 8.2 Programme Simulation

Another distinct purpose of modular architectures is to model programmes, as a means to induce them from data or to simulate symbolic algorithms. The simplest (and least expressive) family of programmes are Finite State Automata (FSA). These receive a neural implementation in the Compositional Recursive Learner (CRL; Chang et al., 2019), similarly to Routing Networks (Rosenbaum et al., 2018) and Modular Networks (Kirsch et al., 2018). In these neural architectures, a loose equivalence can be drawn as follows: transformations induced by modules are transition functions (arcs in the graph), input and output representations are the states (nodes in the graph), and the input is the starting state. A memoryless routing function selects the transition based on the current state. Thus, the programme graph is constructed *dynamically*. The final states are defined as those reached after the router selects a special end-of-computation action.

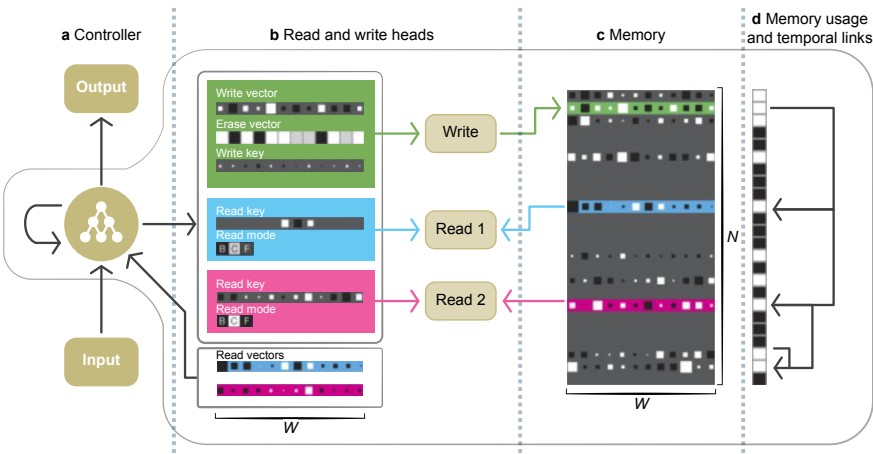

Figure 9: An example of **Programme Simulation** (§ 8.2): Differentiable Neural Computer (Graves et al., 2016). A recurrent neural controller iteratively receives an input from the environment, writes to / reads from memory, and produces an output. Read and write operations are based on attention between generated keys and memory entries. A special mechanism keeps track of memory usage and temporal links between entries.

On the other hand, a programme graph can be constructed *globally* based on the task description before processing the data. In particular, Neural Module Networks (NMNs; Andreas et al., 2016b;a) rely on an off-the-shelf semantic parser (and custom rules) to map a query in natural language into a tree graph. The nodes of this graph are learnable modules characterised by: 1) their *types* of input (raw image features and/or attention scores) and output (attention scores or labels); and 2) the particular *instances* of a type, indicated as an argument in the form of a natural language string. For instance, the module find[cat] takes an image and returns attention scores over the regions that contain cats. Compositionality is achieved by sharing weights across modules with the same type or instance. NMNs have been further extended to be amenable to end-to-end training without the aid of an external parser (Hu et al., 2017). In this case, the mapping from queries to programme graphs is learned by imitating expert demonstrations while the module parameters are learned based on the downstream loss of visual question answering.

In addition to the routing function and computation functions, a model can be extended with an external memory. In fact, these three mirror the fundamental components of a computer architecture: elementary operations, logical flow control, and a random-access memory that can be read and written to (von Neumann, 1945; Graves et al., 2014). While (appropriately wired) recurrent neural networks have been shown to be Turing-complete (Siegelmann & Sontag, 1995), separating the three functions into distinct components provides an inductive bias to simulate the workflow or a computer programme. Neural Turing Machines (NTMs; Graves et al., 2014) introduced a fully differentiable read–write memory matrix that interfaces with the main recurrent network through an attentional mechanism. In particular, this memory can be addressed both based on content (i.e., the match between its entries and the current input) and based on location, in order to store and retrieve temporally ordered information in contiguous entries. NTMs were further extended into the Differentiable Neural Computer (DNCs; Graves et al., 2016, Figure 9), which amended some of the limitations of NMTs, such as avoiding interference in the memory, freeing up previously written locations, and storing temporally ordered sequences in non-contiguous chunks. Another family of memory-augmented methods include the Neural Programmer Interpreter (NPI; Reed & de Freitas, 2016). This model is trained with full supervision from execution traces or through reinforcement learning (Pierrot et al., 2019). In particular, a core recurrent network receives information from a programme module, as well as representations from the environment module. In its output, it produces the index for the next sub-programme and its arguments (as well as a special termination symbol).

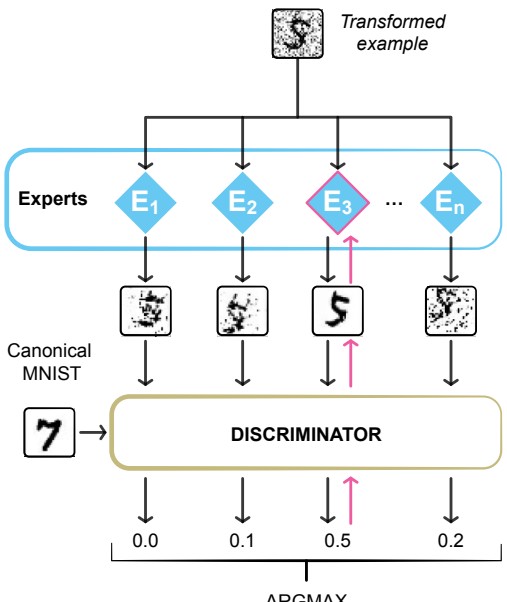

Figure 10: An example of **Causal Inference** (§ 8.3): Causal Independent Mechanisms (Parascandolo et al., 2018). A transformed example is routed to an expert which maps it to the original distribution. An adversarial discriminator attempts to distinguish between reconstructed and original examples.

Finally, a recent thread of research focused on *simulating* the behaviour of symbolic algorithms with vanilla (non-modular) neural networks. An example is neural algorithmic reasoning (Veličković & Blundell, 2021). First, a processor network is trained to emulate the output of a symbolic programme (e.g., Dijkstra's algorithm for shortest paths) that operates on abstract representations (e.g., weighted graphs). Second, encoder and decoder networks can be trained to operate on sensory real-world data while matching the input–output types expected by the processor network.

Among the main applications for programme simulations are settings where sub-problems are shared, such as multi-task or curriculum learning. By distilling the most common functionalities into modules, these can be reused to generalise compositionally to new sequences of sub-tasks. Another application is compositional reasoning, such as (visual) question answering (Andreas et al., 2016b;a). In general, external memory is useful for reasoning over complex data structures, such as graphs (Graves et al., 2014; 2016; Reed & de Freitas, 2016). Finally, neural models can emulate symbolic algorithms to extend their capabilities to operate on sensory real-world data.

### 8.3 Causal Discovery and Inference

Modularity in the design of a model may be assumed to reflect the modularity in the (physical) mechanisms of the world. In fact, a crucial assumption in causal inference (Schölkopf et al., 2012) is that such mechanisms underlying data generation are independent, as they do not influence each other, and reusable, as they may play a role in multiple distributions. Consequently, if one of the mechanisms, which defines a conditional distribution in the model graph, changes—possibly because of an intervention—the other modules remain invariant. If a machine learning model mirrors this modular structure, it is better suited to generalise in a sample-efficient way to new tasks: in fact, local distribution shifts require updating only the corresponding module parameters, which in turn results in faster adaptation (Bengio et al., 2020; Mittal et al., 2022).

The key challenge for this problem is how to specialise each module towards a specific mechanism based uniquely on observational data, especially when the number and nature of the mechanisms are unknown. Competition among the modules through top-$k$ routing (see § 4.2.2) is a common feature of many proposed

solutions.[24] Parascandolo et al. (2018) show how to *invert* causal independent mechanisms through a modular neural architecture, given data from the original distribution and an unlabelled mixture of their transformations (see Figure 10). Their model consists of a mixture of experts and an adversarial discriminator, which enforces that the inverted transformation lies in the support of the original distribution. Another architecture relying on module competition and capable of modelling sequential data is Recurrent Independent Mechanisms (RIMs; Goyal et al., 2021). Here, the modules are recurrent networks with separate parameters, each representing a different transition dynamics. However, their states are not entirely independent, as active modules are allowed to communicate through attention. This reflects a second assumption, namely that the dependencies among variables are highly sparse (Mittal et al., 2022). Attention can also serve to direct the flow of bottom-up and top-down information (Mittal et al., 2020).

Another challenge of neural causal discovery is jointly inducing abstract latent variables (such as objects or entities) from low-level perception (e.g., pixels of an image) while simultaneously learning the causal graph underlying such variables, which determines how they interact (Ke et al., 2021a). The lacklustre abilities of vanilla neural models to understand the compositional properties of symbolic building blocks, i.e. their 'binding problem', arguably explains their current shortfalls in systematic generalisation (Greff et al., 2020). *Object-centric learning* holds promise to mitigate these limitations. For instance, it can be facilitated by slot attention, which is a fully differentiable and iterative attention mechanism that interfaces between perceptual representations and slots, a set of unordered placeholder variables (Locatello et al., 2020). (Didolkar et al., 2021) propose Neural Production Systems, where rule templates can be bound to specific entities present in the working memory, in order to update their representations. In particular, rules are MLP modules and the matching with entities (triggering updates) is parameterised by attention.

Crucially, observational data alone is often[25] insufficient to learn structural causal models as they may not be identifiable (Pearl, 2009). Hence the necessity to augment observation with *interventions* and *counterfactuals*. These allow for answering questions about cause–effect relationships rather than mere correlations. In real-world scenarios, however, the nature and number of interventions are unknown Ke et al. (2021a). In this setting, there is no formal guarantee that causal discovery succeeds. Yet, Ke et al. (2019) finds that DAG discovery on interventional data based on continuous optimisation recovers causal graphs reliably. In particular, modular architectures surpass both vanilla models and graph neural networks (Ke et al., 2021a). Recently, Geffner et al. (2022) perform causal inference in a deep non-linear additive noise structural equation model, based on autoregressive flows. Variational inference is used to learn a posterior over causal graphs. The learned functions can be further used to estimate conditional average treatment effects based on simulations.

The main purpose of these deep modular methods is causal inference and discovery, which has applications in several branches of medicine and economics (Geffner et al., 2022). In addition, these methods are particularly relevant in grounded settings, where the distribution of the observations from the environment changes as the agent learns better policies (Goyal et al., 2021). Moreover, causal discovery can be combined with model-based RL methods to learn a self-supervised model of the environment, i.e. its variables and their causal dependencies, from trajectories of observations, actions, and rewards. This allows for simulating the potential outcomes of a policy before execution and thus estimating better value functions, which dramatically improves sample efficiency in agents (Ke et al., 2021a). Another common application of this family of modular neural architectures is out-of-distribution generalisation: for instance, zero-shot transfer to images of different sizes or sequences of different lengths (Goyal et al., 2021).

---

[24]In addition to causal inference, this strategy is also inspired by the *global workspace theory* (Baars, 2005). This theory postulates specialised modules compete to update a shared workspace, and the resulting communication bottleneck creates a crucial inductive bias in human cognition.

[25]Unless specific assumptions are made about the data generating process, such as linear but non-Gaussian data.

## 9 Conclusions

- **Modularity** is defined as the functional specialisation of the components of a system.
- Specialised sub-networks may emerge in vanilla neural modules (from multitask training or regularisation), but they are seldom reused and recombined.
- Deep modular architectures rest on the separation between **computation** functions on the one hand and **routing** and **aggregation** functions on the other.
- Computation functions may consist of any neural module. Modules may modify the original **parameters**, be concatenated to the **input**, or composed with the original **function**.
- All composition strategies are **equivalent** to summing the original output with a term depending on the new module. In practice, however, they offer different **trade-offs** between efficiency (in time and space, during training and inference) and performance.
- Routing controls the flow of information, i.e., module selection. In **fixed** routing, it is determined *a priori* based on expert knowledge. When this is not available, routing parameters are **learned**.
- Learned routing is challenging because of **training instability**, **module collapse**, and **overfitting**. Thus, learned routing often underperforms fixed routing.
- Routing can be **conditioned** on (parts of) the input or metadata such as task identity. Routing can take place at different **levels**, such as globally for the whole model or layer-wise.
- **Soft** routing assigns every module a continuous score and performs a weighted combination of their outputs. It is amenable to being learned via gradient descent but is highly inefficient.
- **Hard** routing activates only a subset of modules via top-1, top-$k$, or variable-size selection. It is learned via reinforcement learning, evolutionary algorithms, or stochastic re-parameterisation. It corresponds to the principles of conditional computation and information bottleneck in cognition.
- **Hypernetworks** can be interpreted as combining unnormalised routing (task embedding) with modules (generator). They can in turn generate parameters of other modules.
- If routing selects multiple modules, these must be **aggregated** via a function.
- Module parameters or outputs can be **interpolated** for aggregation, according to scores from the routing function, an attention mechanism, or via simple averaging.
- Alternatively, aggregation may involve composing the module functions, either **sequentially** or based on a **tree graph** obtained from global routing.
- The applications include **parameter-efficient fine-tuning** in NLP, computer vision, and speech processing. These rely on the same types of modules and fixed routing. In addition to increased efficiency, this prevents negative interference and enables zero-shot transfer.
- Modularity also serves the purpose of **generalising to new tasks** systematically, by recombining modules and locally updating them.
- Modular deep learning transcends the confines of private research: it enables **community-driven** sharing, expanding, reusing, and updating of the modules.
- In **hierarchical reinforcement learning**, modular **options** serve as abstractions between task goals and low-level actions and observations. They facilitate planning in long-horizon and sparse-reward tasks and increase sample efficiency due to transferability.
- In **programme induction**, the components of deep models can mirror a computer architecture: modules are elementary operations and routing is logical flow control. These are often augmented by an external read–write **memory**. Modules can also simulate symbolic algorithms.
- In **causal discovery and inference**, modules may be taken to correspond to physical mechanisms that are independent and reusable.
- Modular deep learning empowers these traditional applications by learning abstractions (options, programmes, causal graphs) **end-to-end from perceptual stimuli**.

### 9.1 Future Work

While recently modularity has attracted increasing attention in research, there remain many interesting open research questions along the axes of modularity introduced in this survey. We provide an overview of some of these directions for future work.

**Combination of Different Types of Computation Functions**   Existing computation functions (see § 3) are mostly associated with a single category: parameter composition, input composition, or function composition. There are a few exceptions such as compacter (Mahabadi et al., 2021a)—low-rank adapters—which combine multiple types. In general, techniques from parameter composition that incorporate sparsity, a low-rank or structural constraint are agnostic of the form of the module. In practice, this should enable more efficient learning and aggregation.

**Learned Module Structure**   Most modules used in current works share the same architecture, which is reused across different settings. Depending on the skill or knowledge that should be learned, a module may need to be structured differently and might require access to another component or other type of data. In the extreme, a model may require a special-purpose architecture to be able to perform a specific capability (Andreas et al., 2016b). As modules are more widely used, they may benefit from being learned in a more flexible manner, perhaps incorporating ideas from neural architecture search (Negrinho et al., 2019) in a module-specific space of architecture primitives.

**Standardising Modularity Evaluation**   Depending on the dimension studied, modular approaches may be evaluated based on a variety of factors including downstream performance, memory footprint, number of parameters, latency, and compositional generalisation. In order to make progress on modular models in general, evaluation should be standardised. Current evaluation is additionally mainly based on existing datasets that are re-purposed to enable modular evaluation such as by framing them in a zero-shot transfer setting. Future work on modularity evaluation should design forward-looking evaluation benchmarks that are designed to test the capabilities of the next generation of modular models such as assessing the composition of skills and acquisition of new types of reasoning abilities.

**Nature of Modular Representations**   While modular representations have been aggregated and composed, it remains mostly unclear how the inductive bias of a computation function influences the modular representation that is learned. In addition, it remains unclear how computation functions differ on a representation level. Beyond the computation function, it is also unclear how the other dimensions of our taxonomy, i.e., the routing function, the aggregation function, and the training setting influence the nature of the modular representations.

**Hierarchical Modularity**   Current approaches mostly do not differentiate between high-level and low-level skills and how they relate to each other. It might also be possible to designate particular parts of the model or dedicated modules to capture a set of specialised skills or options, and clearly distinguish between other (sets of) skills. At fine-tuning, even more specialised sub-modules could be learned focused only on the previously designated modules. One example might be learning fine-grained specialised subnetworks over larger subnetworks of the original model, offering gradual module specialisation.

**Learned Routing for Pre-training**   Fixed routing (see § 4) is the most common strategy to disentangle knowledge into modular parts of the model. However, fixed routing limits the usability of the proposed methods as they cannot be used on data, which lacks the metadata needed for fixed routing; for instance, when training on heterogeneous data, metadata such as domain information often does not exist. While learned routing methods do not require this metadata to perform routing a priori, they suffer from training difficulties (as discussed in § 4.2). This opens up research directions that enable modular pre-training with learned routing, which would make modular models applicable to a broader set of data.

**Modular Instruction Tuning**   The main way in which current LLMs are specialised to particular downstream settings is via instruction tuning (Wei et al., 2022b), i.e., fine-tuning on a collection of tasks

described via instructions. These tasks are increasingly defined based on a set of skills and capabilities that a model should learn, which opens the room to developing modular instruction tuning methods that enable the acquisition, updating, and composition of specialised knowledge.

**Benchmarking Routing Methods**   Existing studies mainly evaluate routing methods based on performance but do not take into account how different routing strategies influence modular representations. In order to make progress on better routing methods, benchmarks and metrics are necessary that compare routing mechanisms from a modularity perspective across different settings.

**Structured and Sparse Aggregation**   Current aggregation methods (see § 5) combine the information from multiple modular components by applying arithmetic operations such as addition and subtraction across all parameters, which likely includes parameters that should not be modified. Structured or sparse aggregation methods could focus on aggregating information within salient subnetworks or parameter groups, which might make aggregation more efficient and improve out-of-distribution generalisation.

**Learned Aggregation Methods**   Most aggregation methods are based on arithmetic operations. Depending on the nature of the modular information, it may be useful to (non-)linearly transform the representations. More complex domain-specific aggregation methods can be learned in conjunction with the modular representations to enable better generalisation to new settings.

**Merging Modular Models**   In recent work, merging models trained with different settings has led to improved performance (Wortsman et al., 2022, *inter alia*). Rather than requiring separate training runs of a model, a multi-task model can alternatively be trained with modular components that are designed to be merged at a later stage. This potentially allows for an architecture, which can be computationally efficiently trained while covering many modalities.

**Extensible Multi-task Models**   Most approaches in multi-task learning have focused on training dense models, with a key limitation being that models cannot easily be extended to new settings. Focusing on training multi-task models with modular components ensures that the baseline models are much easier to adapt and extend to new settings. Given the trend of pre-training larger and larger models from scratch, modularising parts of such models and developing modular methods that can be shared across different architectures and model sizes may lead to more sustainable model development.

## Acknowledgements

Ivan Vulić is supported by a personal Royal Society University Research Fellowship (no 221137; 2022–).

We are grateful to Colin Raffel for his comments and suggestions, which have greatly improved the manuscript. We thank Andrea Gesmundo for feedback on a draft of this paper. We are thankful to Kyunghyun Cho and Alessandro Sordoni for stimulating discussions. We thank the anonymous reviewers for helpful suggestions and feedback.

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
