# OpenReview forum: "Modular Deep Learning"
_TMLR — Accepted by TMLR_

### Review · Reviewer_5FSf · 2023-07-18

**Summary Of Contributions:**

The paper presents a review of modular deep learning. It presents a unified view of different modular deep learning methods, characterizing them by computation function, routing function, aggregation function, and training setting. For each of these, the authors provide descriptions of the current approaches in the field. Next, the authors discuss applications in transfer learning and other settings such as hierarchical reinforcement learning. Finally, the authors conclude and discuss next steps.

**Audience:**

Yes

**Broader Impact Concerns:**

No broader impact concerns.

**Claims And Evidence:**

Yes

**Requested Changes:**

**Critical**

Page 21 typo:  "in Modular Meta Learning,..."


**Would strengthen**

Please consider making the paper overall more concise (please see "Strengths and Weaknesses" for more details)
Please consider bold font for the + and - in Table 3

**Strengths And Weaknesses:**

**Strengths**

One of the major strengths of the paper is the way the authors unify the many methods of modular deep learning under a single framework. The fact that a single, relatively simple algorithm (Algorithm 1) can describe all these methods is surprising, but it also provides a refreshing and clarifying view of these methods.

The paper is very well written. The authors have done a wonderful job of clearly and carefully describing each method proposed in recent years. Sections 1 - 6 are excellently structured. Figures are well illustrated. Table 2 is quite useful. The choice of notation is appropriate. The boxed conclusion section is helpful.

Overall, I believe this paper could be a strong new reference point for researchers who want an overview of the field of modular deep learning.

**Weaknesses**

As a review paper, the paper doesn't present any technically new content, but this is ok since it doesn't claim to.

In my view, one of the drawbacks of this paper is its length. While the paper is quite comprehensive, it may be more helpful for the reader to read shorter sections that illustrate the key principles rather than a longer section covering details more finely. The authors may wish to consider succinctly describing only 1-2 representative examples of each of the possible methods for each aspect of modularity (e.g. parameter composition for computation function) rather than providing a more comprehensive overview.

I am also uncertain about the value of sections 7 and 8. The previous sections describe how modular deep learning methods are designed, and the structure of the sections appropriately reflects the different aspects of these methods. Sections 7 and 8 describe the usefulness of modular deep learning methods. Unfortunately, there seems to be relatively little underlying structure in how these methods are used, and the structure of these sections reflects this; at certain points, the sections seem to simply list different use cases. I would suggest significantly condensing these sections to briefly describe only a few key applications.

---

> ### Author Response · Authors · 2023-09-15
>
> We want to express our gratitude for your detailed review and valuable feedback. Your comments and suggestions have significantly contributed to improving the quality and comprehensiveness of our work.
>
> We acknowledge your concern regarding the length of the paper. While we understand that the paper covers a wide range of topics, we believe that the comprehensive nature of this survey paper is one of its strengths. Modular deep learning is a rapidly evolving field with numerous approaches and applications, and our aim is to provide a detailed and informative resource for researchers and practitioners.
>
> However, to improve readability and navigation for the readers, we have implemented the following changes:
> - **Table of Contents:** We have added a table of contents at the beginning of the paper to provide an overview of the paper's structure. This will help readers navigate the paper more effectively and find the sections of interest.
> - **Summary Boxes:** We have included brief summary boxes at the beginning of each of the sections 3-6, highlighting the key takeaways from these sections. These boxes serve as TL;DR summaries and allow readers to quickly grasp the main points without delving into the finer details.
>
> Regarding sections 7 and 8: These sections are intended to offer a broad overview of how modular deep learning methods have been applied in specific fields. While they may seem less structured compared to earlier sections, this structure aligns with the nature of applications in various domains, which often involve diverse use cases. Sections 7 and 8 build upon the concepts discussed in the earlier sections and aim to provide direct examples of these concepts in real-world scenarios. We believe that these sections will serve as valuable references for readers interested in those application domains.

---

### Review · Reviewer_eFvt · 2023-08-04

**Summary Of Contributions:**

This paper
* Presents a survey of modular deep learning techniques in the literature
* Unifies a variety of techniques under a single function
* Compares different modular techniques in terms of axes like parameter efficiency, training efficiency and inference efficiency


**Audience:**

Yes

**Broader Impact Concerns:**

I don't think there are any ethical concerns with this work.

**Claims And Evidence:**

Yes

**Requested Changes:**

See above.

**Strengths And Weaknesses:**

Strengths:
* I think this paper will be really helpful for the wider community to understand a burgeoning research direction.
* The paper is very well written
* The unification of different techniques under the same functional form is quite insightful!
* The covered content is vast in scope, and quite a comprehensive survey.

Weaknesses:
All the following weaknesses will only make the paper better and are not critical for securing my recommendation for acceptance.

* The authors could mention BASE layers (Lewis et al 2021) as a specific technique to prevent expert collapse in Section 4.2.1
* The authors could mention Shen et. al 2023 as another mechnanism to improve MoE finetuning approaches in Section 4.2.1
* There is an interplay between aggregation techniques and sparsity which is not mentioned; if the modular model is sparse, ie only a few experts are active at inference time, one can get away with fewer active modules with ensembling at the same inference costs as other techniques like merging.
* The authors could compare the inference costs of various aggregation techniques, ie that representation aggregation is necessarily more expensive than parameter aggregation (the latter of which reduces the cost of inference to a single model), but that representation aggregation techniques tend to outperform parameter averaging.
* I think the paper sections would benefit from a similar summary box that you have at the end of the paper, since the content is so detailed it can be hard to keep track of the general takeaways from each section.
* At each section, I would appreciate tables that the presented presented techniques across the main axes explored (performance inference costs, etc)
* Compositionality is only briefly mentioned, but I think a key direction going forward. Can we compose different skills from specialized models towards new tasks? Joel et al 2023 suggests that it might be possible.
* Additional merging techniques that could be mentioned:
- dataless knowledge fusion: https://arxiv.org/abs/2212.09849
- TIES merging - https://arxiv.org/abs/2306.01708
- stochastic weight averaging - https://arxiv.org/abs/2001.02312
- model fusing - https://arxiv.org/abs/2204.03044

Additional references:
Lewis et al 2021 - ​​https://arxiv.org/abs/2103.16716
Shen et al 2023 - https://arxiv.org/abs/2305.14705
Jang et al 2023 - https://arxiv.org/abs/2302.03202

---

> ### Author Response · Authors · 2023-09-15
>
> We greatly appreciate your thorough review and the constructive feedback. Your comments and suggestions have been instrumental in improving the quality and comprehensiveness of our work. Below, we address each of your concerns and describe the actions taken in response:
>
> > ... BASE layers...
>
> We appreciate the reviewer's suggestion regarding BASE layers. In the original paper, we indeed addressed BASE layers in Section 4.2.3, where we discussed their role in token-level routing algorithms as a means of load balancing. However, we want to emphasize that BASE layers are a specific technique tailored to token-level routing and primarily aimed at mitigating expert collapse in that context. While BASE layers are a valuable contribution to the field, they are not a universally applicable solution for all methods of learning to route. Therefore, we believe that our original placement of the discussion in Section 4.2.3, which delves into token-level routing, is appropriate given the specificity and scope of BASE layers.
>
> > ... aggregation techniques and sparsity ...
>
> We have added a new section to the introduction of the Routing section, discussing the efficiency of routing algorithms. However, we aim to provide a comprehensive survey of modular neural networks, with a primary emphasis on their functional aspects. While discussing the interplay between aggregation techniques and sparsity is valuable, it is important to note that it is not the central focus of our paper. However, we have added footnotes to emphasize this aspect as an additional benefit of sparse modular networks.
>
> > ... inference costs...
>
> We would like to clarify that efficiency is not the central focus of our work. While we acknowledge that efficiency is indeed an essential aspect of modular architectures and that it plays a crucial role in real-world applications, it is challenging to provide a comprehensive comparison of efficiency across various aggregation techniques. Measuring efficiency encompasses multiple factors such as parameters, FLOPs (floating-point operations), training time, inference time, and resource utilization, and these factors may have different degrees of importance in different scenarios.
>
> Conducting a thorough and fair comparison of efficiency for different aggregation techniques is a complex undertaking that requires dedicated research efforts and specific experimental setups tailored to address this aspect comprehensively. Due to the scope of our paper, which primarily focuses on the taxonomy, comparison, and discussion of various modular neural network components, we are unable to delve deeply into efficiency comparisons.
>
> We believe that addressing efficiency-related questions is an important and valuable area for future research, but it is beyond the current scope of our work. We encourage future investigations to explore and provide detailed insights into the efficiency aspects of modular architectures.
>
> > ... summary box ...
>
> We appreciate your suggestion for summary boxes at the end of each section. To provide readers with clearer takeaways, we have integrated summary boxes for Sections 3-6.
>
> > ... tables ..
> We already provide tables for presented techniques in the Computation Function section, and we aim to integrate similar tables for the Routing and Aggregation sections in the camera-ready version.
>
> > ...Compositionality...
>
> We agree with importance of compositionality in modular architectures. Indeed, the concept of compositionality is a central theme in the field of modular neural networks, and it plays a significant role in enabling models to generalize across diverse tasks and exhibit positive transfer effects.
> Our paper extensively discusses the notion of compositionality in the "Aggregation" section, which explores various methods for combining or aggregating information from different specialized modules. Compositionality can be viewed as an inherent property of aggregation techniques, and it is a crucial aspect of modular architectures.
> As the reviewer correctly notes, compositionality allows for the assembly of different skills or capabilities from specialized models into new tasks or domains. For example, if we have modules that excel at processing one language and modules that are proficient in named entity recognition (NER), we can compose them to perform NER in a specific language, showcasing the versatility of modular approaches.
> Furthermore, we acknowledge that the mentioned paper by Joel et al. in 2023 underscores the significance of aggregation and composition of modules in the era of Large Language Models (LLMs). These methods continue to demonstrate a high impact on downstream performance and are essential for pushing the boundaries of what LLMs can achieve.
>
> > missing references
>
> We appreciate your suggestions for additional merging techniques. We have included these references and discuss them in the relevant sections to provide a more comprehensive overview.

---

### Review · Reviewer_oaBm · 2023-08-15

**Summary Of Contributions:**

This comprehensive study delves into modular designs in deep neural networks, providing a systematic taxonomy of different components and offering a broad view of their applications. The authors present detailed comparisons between various methods, suggestions for modular designs, and insightful future directions.



**Audience:**

Yes

**Claims And Evidence:**

Yes

**Requested Changes:**

N/A

**Strengths And Weaknesses:**

## Strengths

- The taxonomy of modular networks is well-defined, with each part explained thoroughly and insightful connections made between them. The authors also shed light on the relationship between hyper networks and these four parts, enhancing understanding.
- It provides a wide-ranging perspective on modular applications like Hierarchical Reinforcement Learning, Programme Simulation, and Causal Discovery and Inference, going beyond common tasks such as Transfer Learning.
- The use of simple yet intuitive examples is commendable. For example, in 7.1.1, the Machine Translation, the authors demonstrate the power of modularity through multilingual NMT adapters.
- The authors also offer suitable suggestions for utilizing and implementing modular networks. They emphasize that while modularity is a good principle, the level and scope of modularity should be considered accordingly (as highlighted in sections 4.3 and 7.1.1).

## Weaknesses

- It would be helpful to include a catalog at the beginning, allowing readers to easily refer to specific parts and form a hierarchical understanding of the work.
- In the Function Composition section (3.3), the reviewer suggests discussing semi-parametric modules [1] [2], which extend the base model with additional components like knowledge bases. Moreover, semi-parametric modules like retrievers also relate to Knowledge Injection in section 7.1.4.
- The Soft Learned Routing section (4.2.3) could benefit from more comprehensive coverage. While different levels of routing and auxiliary information are addressed, it would be valuable to discuss loss [3] and architecture designs [4] that facilitate learning to route while encouraging module specialization (as mentioned briefly in 4.2.1).

[1] Verga, P., Sun, H., Soares, L. B., & Cohen, W. (2021, June). Adaptable and interpretable neural memory over symbolic knowledge. In _Proceedings of the 2021 conference of the north american chapter of the association for computational linguistics: human language technologies_ (pp. 3678-3691).

[2] Cheng, X., Lin, Y., Chen, X., Zhao, D., & Yan, R. (2023). Decouple knowledge from paramters for plug-and-play language modeling. _arXiv preprint arXiv:2305.11564_.

[3] Shen, Y., Zhang, Z., Cao, T., Tan, S., Chen, Z., & Gan, C. (2023). ModuleFormer: Learning Modular Large Language Models From Uncurated Data. _arXiv preprint arXiv:2306.04640_.

[4] Chi, Z., Dong, L., Huang, S., Dai, D., Ma, S., Patra, B., ... & Wei, F. (2022). On the representation collapse of sparse mixture of experts. _Advances in Neural Information Processing Systems_, _35_, 34600-34613.

---

> ### Author Response · Authors · 2023-09-15
>
> We would like to express our gratitude for your thoughtful and constructive feedback which has been instrumental in improving the quality and comprehensiveness of our work. Below, we address your concerns and provide a summary of the changes made to the manuscript:
>
> > It would be helpful to include a catalog at the beginning, allowing readers to easily refer to specific parts and form a hierarchical understanding of the work.
>
> To enhance the accessibility of the paper, we have added a table of contents at the beginning. This provides readers with a clear roadmap of the paper's structure, allowing for easier reference and navigation.
>
> > In the Function Composition section (3.3), the reviewer suggests discussing semi-parametric modules [1] [2], which extend the base model with additional components like knowledge bases. Moreover, semi-parametric modules like retrievers also relate to Knowledge Injection in section 7.1.4.
>
> We appreciate the suggestion to discuss semi-parametric modules.  We have now extended sections (3.3), (5.2), and (7.1.4). We incorporate a discussion of semi-parametric modules and related methods that extend base models with additional components, such as knowledge bases. We further explore the utility of input aggregation methods in retrieval-augmented generation, which includes a discussion of how external retrieval models can be employed to retrieve external knowledge for knowledge-intensive NLP tasks, aligning with the broader theme of knowledge enhancement within modular neural architectures.
>
> > The Soft Learned Routing section (4.2.3) could benefit from more comprehensive coverage. While different levels of routing and auxiliary information are addressed, it would be valuable to discuss loss [3] and architecture designs [4] that facilitate learning to route while encouraging module specialization (as mentioned briefly in 4.2.1).
>
> We have begun to address the need for more comprehensive coverage in the Soft Learned Routing section (4.2.3). This section now includes discussions on loss functions and architecture designs that facilitate learning to route while encouraging module specialization. We have also extended this section to address MoE methods that mitigate module collapse, with further enhancements planned for the camera-ready version.
>
> Thank you once again for your time and expertise in reviewing our paper. We look forward to any additional feedback or suggestions you may have

---

### Author Response · Authors · 2023-09-15
**General Response**

We want to express our sincere gratitude to all the reviewers for their thorough and insightful feedback, which has significantly contributed to enhancing the quality and comprehensiveness of our work. We greatly value your time and expertise in reviewing our paper. Below, we provide a summary of the key changes made in response to your comments and address your concerns:

Changes Made in Response to Reviewers:

- **Table of Contents:** We have added a table of contents at the beginning of the paper to improve the paper's structure and assist readers in navigating its content effectively.
- **Summary Boxes:** To enhance readability and provide clear takeaways, we have included brief summary boxes at the end of sections 3-6, highlighting key points for each section.
- **Extended Discussion on RAG:** We have extended sections (3.3), (5.2), and (7.1.4) to discuss retrieval augmented and related methods that extend base models with additional components, such as knowledge bases.
- **Incorporated Additional References:** We have included additional references suggested by the reviewers to provide a more comprehensive overview of the field.

We believe that these changes address the reviewers' concerns and enhance the overall quality and accessibility of our paper. Once again, we thank you for your valuable feedback, and we look forward to any additional comments or suggestions you may have.

---

### Decision · Action_Editor_uErd · 2023-11-04

**Recommendation:** Accept as is

**Comment:**

All the reviewers found the paper to be well-written and a valuable contribution. The authors have incorporated suggestions including additional discussion, a table of contents and summary boxes to further strengthen the paper.

**Audience:**

This paper will considerably inform future work into modular architectures and will be of broad interested to the TMLR audience.

**Claims And Evidence:**

This work presents a survey of modular architectures for deep learning, with an emphasis on providing a systematic taxonomy of different techniques as well as comparing them across different axes (like parameter efficiency, training efficiency and inference efficiency). The paper is clear and well-written.